# Improving mass spectrometry analysis of protein structures with arginine-selective chemical cross-linkers

Alexander X. Jones[1,8], Yong Cao [2,3,8], Yu-Liang Tang[1,8], Jian-Hua Wang[3], Yue-He Ding[3], Hui Tan[1], Zhen-Lin Chen[4,5], Run-Qian Fang[4,5], Jili Yin[4,5], Rong-Chang Chen[5,6], Xing Zhu[5,6], Yang She[3], Niu Huang [3], Feng Shao [3], Keqiong Ye [5,6], Rui-Xiang Sun[3], Si-Min He [4,5], Xiaoguang Lei [1] & Meng-Qiu Dong [3,7]

Chemical cross-linking of proteins coupled with mass spectrometry analysis (CXMS) is widely used to study protein-protein interactions (PPI), protein structures, and even protein dynamics. However, structural information provided by CXMS is still limited, partly because most CXMS experiments use lysine-lysine (K-K) cross-linkers. Although superb in selectivity and reactivity, they are ineffective for lysine deficient regions. Herein, we develop aromatic glyoxal cross-linkers (ArGOs) for arginine-arginine (R-R) cross-linking and the lysine-arginine (K-R) cross-linker KArGO. The R-R or K-R cross-links generated by ArGO or KArGO fit well with protein crystal structures and provide information not attainable by K-K cross-links. KArGO, in particular, is highly valuable for CXMS, with robust performance on a variety of samples including a kinase and two multi-protein complexes. In the case of the CNGP complex, KArGO cross-links covered as much of the PPI interface as R-R and K-K cross-links combined and improved the accuracy of Rosetta docking substantially.

[1] Beijing National Laboratory for Molecular Sciences, Key Laboratory of Bioorganic Chemistry and Molecular Engineering of Ministry of Education, Department of Chemical Biology, College of Chemistry and Molecular Engineering, Synthetic and Functional Biomolecules Center, and Peking-Tsinghua Center for Life Sciences, Peking University, 100871 Beijing, China. [2] School of Life Sciences, Peking University, 100871 Beijing, China. [3] National Institute of Biological Sciences (NIBS), 102206 Beijing, China. [4] Key Lab of Intelligent Information Processing, Chinese Academy of Sciences (CAS), Institute of Computing Technology, CAS, 100049 Beijing, China. [5] University of Chinese Academy of Sciences, 100049 Beijing, China. [6] Key Laboratory of RNA Biology, CAS Center for Excellence in Biomacromolecules, Institute of Biophysics, Chinese Academy of Sciences, 100101 Beijing, China. [7] Tsinghua Institute of Multidisciplinary Biomedical Research, Tsinghua University, 102206 Beijing, China. [8] These authors contributed equally: Alexander X. Jones, Yong Cao, Yu-Liang Tang. Correspondence and requests for materials should be addressed to X.L. (email: xglei@pku.edu.cn) or to M.-Q.D. (email: dongmengqiu@nibs.ac.cn)

protein rarely acts alone; it interacts with other proteins either stably or transiently to execute biological functions and to allow regulation. Characterizing protein–protein interactions (PPI) is essential to understanding the biology of the cell. Chemical cross-linking of proteins coupled with mass spectrometry (CXMS) has emerged as a powerful technique for obtaining structural information of proteins and protein complexes[1–6]. The general workflow of CXMS involves four steps[7]: (1) cross-linking under mild conditions that maintain the native conformations of proteins and protein complexes; (2) protease digestion, which generates cross-linked peptide pairs along with linear peptides that are either unmodified, mono-, or loop-linked by the cross-linker; (3) liquid chromatography-tandem mass spectrometry (LC-MS/MS) analysis of this peptide mixture; (4) identification of cross-linked peptides from the mass spectra and mapping of the cross-linked amino-acid residues, in pairs, in the parent proteins[8–10]. By revealing spatial proximity between cross-linked residues in a very convenient way, CXMS complements NMR, crystallography, and cryo-electron microscopy (cryo-EM) to locate a polypeptide in a protein or a protein subunit within a complex[11–15]. CXMS is particularly helpful in characterizing flexible regions or dynamic interactions for which high-resolution data are often not available[16–18]. CXMS can be used alone to map approximately the interacting regions between proteins[19,20], or to detect large conformational changes[21].

In theory, non-selective cross-linkers such as glutaraldehyde and photo-induced cross-linkers can retrieve more structural information than residue-selective cross-linkers, because they can react with most if not all amino-acid residues. However, identification of the resulting cross-linked peptides is a huge challenge due to an explosion of the search space and a lack of proper evaluation of identification results.

Up to now, residue-selective chemistry is essential to generate well-defined products for facile identification of cross-linked peptides[22–24]. Present CXMS analyses predominantly rely on lysine-targeting cross-linkers that use N-hydroxysuccinimide (NHS) esters as reactive groups[25]. Other cross-linking reagents developed so far have only limited use for various reasons. For example, the low prevalence of cysteines in proteins (1.4% in the UniProtKB/Swiss-Prot database) discourages the use of thiol specific cross-linkers[26]. Aspartate and glutamate residues are abundant in proteins, but the carboxyl selective cross-linking reagents—either bifunctional carbonyl hydrazides with a coupling reagent[27,28] or bifunctional diazo compounds without a coupling reagent[29]—are not yet efficient enough to produce a large number of cross-links. The zero-length cross-linkers EDC/NHS or DMTMM[28] facilitate an aspartate or glutamate residue to form an amide bond with a nearby lysine residue that is supposedly less than 9.7 or 11 Å away (Cα-Cα, D/E-K), but the vast majority of the D/E-K cross-links obtained exceed this distance limit when inspected using protein crystal structure models[30].

Because of the nearly exclusive use of NHS-ester cross-linkers such as bis(sulfosuccinimidyl)suberate (BS³), disuccinimidyl suberate (DSS), and DSSO[31], CXMS may or may not be successful depending on the number and position of lysine residues in a protein complex. Not surprisingly, the spatial information retrieved by CXMS is sometimes not enough to locate the PPI interface. A survey of 1808 protein complexes in the PDB database showed that around 40% would give less than five inter-molecular cross-links using BS³ or DSS, and 20% would give none[30]. There is a pressing need for new cross-linkers to improve the structural coverage of CXMS.

Arginine is an abundant (5.1%) amino acid and plays an important role in protein structure and function. Due to the strongly basic guanidine group ($pK_a \sim 12$), it is always protonated under physiological conditions, and is thus solvent-exposed to allow H-bonding stabilization with water. It frequently serves as a recognition site for other proteins or nucleic acids by electrostatic interaction with negatively charged carboxylate residues or the phosphate moieties in DNA or RNA[32]. It is the second most enriched amino acid in protein–protein interaction hot spots (after tryptophan)[33]. Therefore, CXMS targeting arginine residues could yield valuable information about PPI surfaces.

Arginine residues are modified non-enzymatically in cells in the presence of highly reactive α-dicarbonyl reagents generated via the Maillard reaction from reducing sugars[34,35]. Laboratory methods for arginine modification also make use of α-dicarbonyl reagents, among which phenyl glyoxals have shown promising reactivity[36–39]. In a proof-of-concept study, Fabris and co-workers used bifunctional aromatic glyoxals PDG and BDG, each with a rigid spacer arm, to cross-link pairs of arginine residues that are spaced out at just the right distance in proteins[40]. However, despite the initial advance, arginine-selective CXMS remains poorly developed. For the three tested proteins, the number of arginine pairs cross-linked by PDG or BDG ranged merely from zero to four[40] and to the best of our knowledge, these cross-linkers have not been used again since 2008. Problems with the arginine-glyoxal reactions include formation of side products and slow degradation of products under basic conditions[36].

In this study, we develop a series of bifunctional aromatic glyoxal cross-linkers (ArGOs) for arginine-selective CXMS. After extensive optimization, we ameliorate the problems of arginine-glyoxal reactions to an extent that some of the ArGOs are now ready to be used in CXMS experiments (vide infra). The best one, ArGO2, generates 10–80 cross-linked arginine pairs for each of the model proteins tested. We also introduce KArGO, an arginine-lysine hetero-bifunctional cross-linker. KArGO has an aromatic glyoxal group paired with an amine-specific, non-NHS ester reactive group. Having access to the combined abundance of lysine and arginine residues, KArGO offers significantly more protein surface coverage than ArGO1/2 and lysine–lysine cross-linkers. KArGO is a low-dosage and rapid-acting cross-linker with particular benefits in characterizing PPI surfaces. Using biologically relevant multi-subunit complexes, we show that ArGO and KArGO both provide structural information that are inaccessible to DSS. Using the yeast H/ACA complex, we show the distance restraints obtained from KArGO and ArGO cross-linking greatly facilitate Rosetta docking of protein–protein interactions. We also demonstrate the utility of KArGO cross-linking in locating the binding interface between two domains of alpha-kinase 1 (ALPK1), and in locating the protein–protein interacting regions within a five-subunit RNA chaperone complex UtpA.

## Results

**Establishment of ArGO cross-linking chemistry.** In order to gauge the potential advantages of arginine-selective cross-linkers for structural biology, we analysed the number of theoretical cross-linkable Arg–Arg and Lys–Arg pairs across 1808 PDB complexes, using a method described before[30]. By combining R–R and K–R cross-links with K–K cross-links, the percentage of protein complexes with ≥5 virtual inter-molecular cross-links increased from 67 to 88% (Supplementary Fig. 1a). Furthermore, K–R cross-linking alone is theoretically more prolific than K–K cross-linking with respect to generating ≥5 virtual inter-molecular cross-links for a given complex (Supplementary Fig. 1b). These results suggest that arginine-selective reagents would be a useful addition to the cross-linking toolbox.

For the development of arginine-selective conjugation, we initially screened several candidate molecules, including

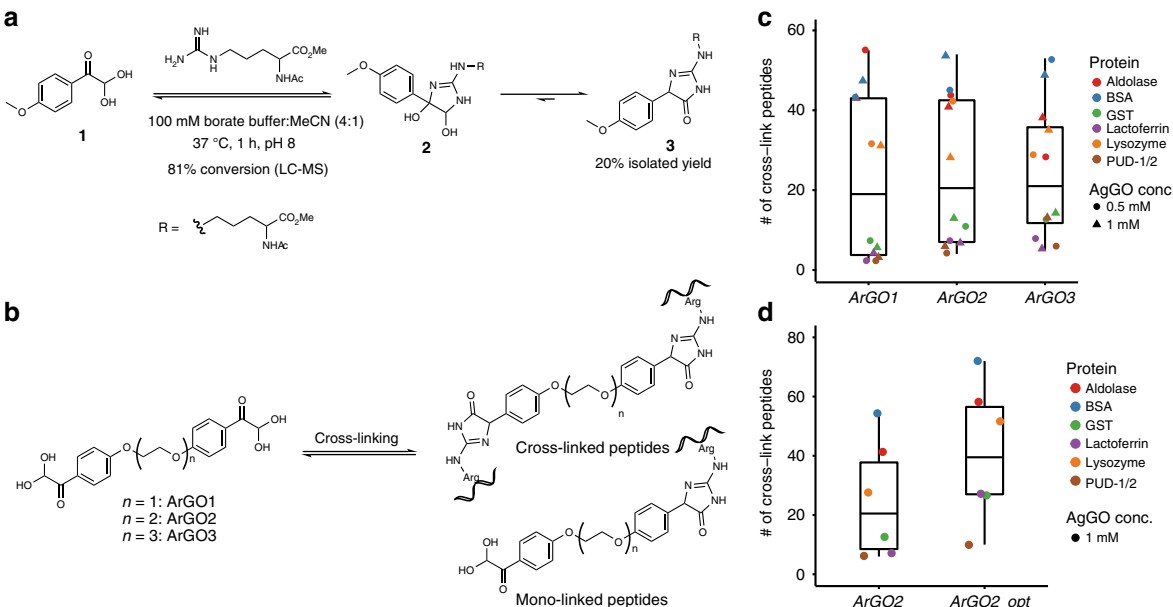

**Fig. 1** Development of ArGO reagents for arginine-arginine cross-linking. **a** Reaction of *p*-OMe-phenyl glyoxal with *N*-acetyl arginine methyl ester. The major product dihydroimidazolone **3** forms following dehydration of the intermediate dihydroxyimidazoline **2**. **b** Structures of ArGO1-3 compounds and of two major cross-linking products. **c, d** Evaluating the performance of ArGO1-3 with six model proteins. **c** Number of identified peptide pairs cross-linked by the indicated ArGO at 0.5 and 1.0 mM for each model protein. Box plots indicating the median (black line), interquartile range (box, the middle 50%), the lowest and the highest number (whiskers) of cross-linked peptide pairs across the proteins shown. The number of cross-link peptides in ArGO1-3 on six model proteins are provided in source data file. **d** Comparison of the number of identified peptide pairs cross-linked by ArGO2 for each model protein under initial (left) and optimized (right) cross-linking conditions. Box plots indicating the median (black line), interquartile range (box, the middle 50%), the lowest and the highest number (whiskers) of cross-linked peptide pairs across the proteins shown. The number of cross-link peptides in ArGO2 and ArGO2 under the optimal conditions on six model proteins are provided in the source data file

fluorotropolone[41], cyclohexanedione[42], and aryl glyoxals for their reactivity towards the guanidine group of *N*α-acetyl arginine methyl ester (Supplementary Fig. 2). *p*-OMe phenyl glyoxal (*p*-OMe-PhGO) **1** in borate buffer emerged as a promising reagent/buffer combination due to high conversion and fewer by-products than other reagents. Borate stabilises the intermediate dihydroxyimidazoline **2** formed by conjugation of arginine with glyoxals, and also activates electron-rich glyoxals in particular towards attack by arginine (Fig. 1a)[39]. The pseudo first-order rate constant for *p*-OMe-PhGO consumption was calculated to be $1.3 \times 10^{-3}$ s$^{-1}$ (Supplementary Fig. 3), comparable with other electron-rich phenyl glyoxals in borate buffer[39]. Among the products formed, which are interconvertible and thus exist in equilibrium (Supplementary Fig. 4)[36], 3,5-dihydro-4-imidazolone **3** is the major and stable one and could be purified chromatographically in 20% isolated yield (Fig. 1a).

Based on the reaction of *p*-OMe-PhGO with the arginine side chain, we synthesized three aromatic glyoxal cross-linkers (ArGO1-3) (Fig. 1b), which possess flexible and electron-donating poly-ethylene glycol (PEG) spacer arms of increasing length to provide different distance restraints in CXMS experiments. The mono-, loop-, and cross-linked peptides—also known as dead-end, intrapeptide, and interpeptide cross-links[43,44]—of ArGO1-3 have mass additions shown in Supplementary Table 1. The arginine selectivity of aromatic glyoxals[36,45] were evaluated using seven synthetic peptides covering all the canonical amino acids (Supplementary Note 1, Supplementary Table 5, Supplementary Table 6, Supplementary Fig. 14). The reaction products were analysed by LC-MS/MS, confirming that adduct **3** (Fig. 1a) is the major product. Side products from peptide *N*-terminal modification or non-covalent conjugation could be greatly reduced after TCEP treatment at 56 °C for 10 min, which is

routinely done during sample preparation to reduce protein disulfide bonds.

Next, using the BSA protein as a model, we systematically optimized the cross-linking conditions for ArGO (Supplementary Fig. 5) and found that ArGO worked well under mild conditions: 50–100 mM borate buffer, pH 7.0–8.0, 0.25–1.0 mM ArGO, 0.6 mg/ml BSA, 30–60 min at 25 °C. We applied these conditions to BSA and five additional proteins (Fig. 1c). Of these, aldolase, BSA, and GST can homo-dimerize or homo-tetramerize; PUD-1/2 is a heterodimer; lactoferrin and lysozyme are monomers. Expected dimer or tetramer bands were clearly detected on SDS-PAGE after cross-linking of aldolase, BSA, GST, and PUD-1/2 by ArGO1-3 (Supplementary Fig. 6). Following LC-MS/MS, we used pLink[8,46] to analyse the CXMS data and found that ArGO1-3 performed with similar efficiency on each of these proteins (Fig. 1c), producing a median of ~20 identified cross-linked peptide pairs per protein. The maximal C$_α$–C$_α$ distance restraints for ArGO1-3 were calculated by molecular dynamics simulation to be 29.4, 33.7, and 37.7 Å, respectively. Using the crystal structures of the six model proteins above, we calculated the C$_α$–C$_α$ distance of each identified cross-link and found that 84% or more of the cross-links were within the distance limit of ArGO1-3 (Supplementary Fig. 7). Hence, the structural compatibility rates of ArGO1-3 compare favorably with those of lysine cross-linkers such as BS[3] and DSS, which are around 80%[30]. As the spacer arms of ArGO1-3 increase sequentially, the number of cross-links produced were expected to increase in the same order, but this was not the case[47]. A similar observation was made in a previous study on lysine cross-linkers, in which EGS—whose spacer arm is longer than BS[3] or DSS—did not produce longer cross-links because a fully extended EGS is energetically unfavourable[30]. Notably, the number of cross-linked peptides identified seems to depend

mainly on the protein: 30–60 cross-links were typically obtained from aldolase, BSA, or lysozyme, well above the number of cross-links (<20) obtained from GST, lactoferrin, or PUD-1/2. The reason for such difference is not entirely clear, but probably involves multiple factors including the number of solvent-exposed arginine pairs at a distance that could be bridged by ArGO, and whether or not an arginine forms a strong salt bridge, which would hinder the arginine-ArGO reaction.

To increase the number of cross-links and make ArGO generally applicable to all proteins, we explored several possibilities. First, we tried to drive the cross-linking reaction of ArGO towards a single product using sodium periodate treatment (Supplementary Fig. 8a)[48], which, according to initial experiments, could oxidatively cleave dihydroxyimidazoline **2** to form a stable N-carbamimidoylbenzamide **A2**. However, this was not satisfactory when tested on a model peptide, for it produced various side products including formylation of the peptide N-terminus (Supplementary Fig. 8b). We then modified the ArGO1 and ArGO2 cross-linkers, altering the position of the glyoxal group relative to the spacer arm on the benzene ring, altering the stereo-electronic properties of the benzene ring, as well as the hydrophilicity, length, and the flexibility of the spacer arm (Supplementary Fig. 9). None of these alternative designs produced more cross-linked peptides than ArGO1 and ArGO2. During this process, we tested more conditions and found that ArGO cross-linking achieved the best results in a buffer containing 50 mM HEPES and 50 mM borate, pH 7.0–8.0 (Supplementary Fig. 10). Presumably, the addition of HEPES improves buffering capacity. We find this dual buffer system to be convenient also because HEPES is a standard buffer for lysine-specific cross-linking using NHS esters such as DSS. If ArGO cross-linking is to be carried out in parallel, there is no need for buffer change—just a supplement of borate from a concentrated stock solution. Lastly, digesting the protein sample cross-linked by ArGO with more trypsin in less time (at protein:trypsin ratio of 50:1 for 4 h instead of 100:1 for overnight) doubled the number of cross-link identifications (Supplementary Fig. 10b), presumably benefitting from less reversion of the major cross-linking product **3** to starting materials (Fig. 1a). In keeping with this, we store ArGO cross-linked samples at −80 °C for no more than a week. Under this optimized condition (Supplementary Fig. 10c), the median number of ArGO2 cross-linked peptides identified from six model proteins increased from 20 to 40 (Fig. 1d).

### Application of ArGO to structural modeling

Having established that ArGOs work well with standard proteins, we next aimed to determine their efficiency with more complex, biologically relevant systems.

Box H/ACA ribonucleoprotein particles (RNP) mediate formation of the pseudouridine post-transcriptional modification at specific sites of rRNAs and small nuclear RNAs[49]. The crystal structure of an archaeal H/ACA RNP, composed of Cbf5 (C), Nop10 (N), Gar1 (G), L7Ae (homologous with Nhp2 (P) in yeast), and the guide RNA has been solved. In 2011, Li et al. reported the structure of the yeast CNG sub-complex (PDB ID: 3u28)[50], and the structure of Nhp2 has been solved separately by NMR[51].

We treated a yeast CNGP complex with ArGO1, ArGO2, and for comparison, BDG[40] (Fig. 2). The result shows that ArGOs are much more effective than BDG (Fig. 2a). An annotated MS/MS spectrum of a representative ArGO cross-linked peptide pair is shown in Fig. 2b. Of the 34 intra-molecular and 21 inter-molecular peptide pairs cross-linked by either ArGO1 or ArGO2, 82% are consistent with the crystal structure. Previously we have performed CXMS analysis of the same complex using DSS and BS3[30]. The combination of ArGO1/ArGO2 and DSS/BS3

cross-linking results markedly increased the protein surface coverage, and there was a high degree of complementarity between these two sets of orthogonal cross-linkers. This was particularly evident at the interface of Nop10(N)- Cbf5(C) (Fig. 2c). DSS provided three cross-links to position the N-terminal region of Nop10 atop the β-sheet region of Cbf5, but little information about the C-terminal half of Nop10, while ArGO1 provided three cross-links connecting the C-terminal half of Nop10 to the α-domain of Cbf5. In the subsequent Rosetta docking[52–54] trials of Nop10 to Cbf5/Gar1 (Fig. 2d–f), incorporating only the DSS restraints resulted in few conformational clusters of a large size (i.e. containing many conformations that are similar to one another) and a relatively large root mean square deviation (RMSD) values with reference to the native structure (Fig. 2d, e). With additional help from the three ArGO1 cross-links, docking results obtained using Rosetta 3.10 with the relax protocol for pretreatment clearly converged better, showing an increase in cluster size and a decrease in RMSD, indicating that these conformational clusters more closely resembled the native structure (Fig. 2d). After refinement, the three largest clusters from DSS only yielded conformations that still deviate significantly from the crystal structure (Fig. 2d, e). After global docking assisted by DSS plus ArGO1 cross-links, the largest conformational cluster is already very close to the native conformation (green, Fig. 2f) as shown by the representative conformer from this cluster (orange, Fig. 2f). By comparison, the representative conformer from the largest cluster obtained with DSS cross-links alone (yellow, Fig. 2f) clearly deviates from the native conformation. Similar results were obtained using Rosetta 3.5 with the prepack protocol for pretreatment (Supplementary Fig. 11). These results validate that the CXMS restraints provided by ArGO greatly improve the accuracy of protein–protein docking.

### Development of the lysine–arginine cross-linker KArGO

Following the R–R cross-linking reagent, we set out to develop a K–R cross-linking reagent by combining the arginine-selective aromatic glyoxal with a lysine selective reactive group. First, we paired ArGO with a classic NHS ester group (Fig. 3a, b), but the resulting hetero-bifunctional cross-linker could not be efficiently synthesized, hampered by extensive hydrolysis of the NHS ester during purification. Turning our attention elsewhere, we discovered ortho-phthalaldehyde (OPA), whose derivatives have been used to conjugate free amines in proteins to produce phthalimidine products[55,56]. OPA is non-hydrolysable[57,58], avoiding problem of linker degradation, and has greater chemoselectivity than NHS esters[55]. OPA is also specific for primary over secondary amines but has not been applied to cross-linking. Synthesis of the OPA–ArGO hetero-bifunctional lysine–arginine cross-linker, which we named KArGO (Fig. 3b), was successful. For cross-linked peptides, the linker mass of KArGO is 334.084 Da (Supplementary Table 1). After optimizing the reaction conditions of KArGO on BSA, we were pleased to see that KArGO is a low-dosage (0.1–0.2 mM) and fast–acting (10–20 min) cross-linker (Supplementary Fig. 12a, b). In addition to trypsin digestion, a separate digestion with trypsin and Asp-N both was conducted concurrently[59]. The trypsin/Asp-N digestion (Supplementary Fig. 12c) increased cross-link identifications significantly (Supplementary Fig. 12d), probably because KArGO-conjugated K and R residues are no longer cleavable by trypsin but these trypsin resistant regions can be cut by Asp-N, which cleaves the peptide bond N-terminal to an aspartic acid residue. KArGO-treated protein samples can be stored as acetone precipitates for at least a week (Supplementary Fig. 12e).

Tested on the six model proteins, KArGO clearly produced more cross-linked peptide pairs than ArGO (Fig. 3c). The lowest,

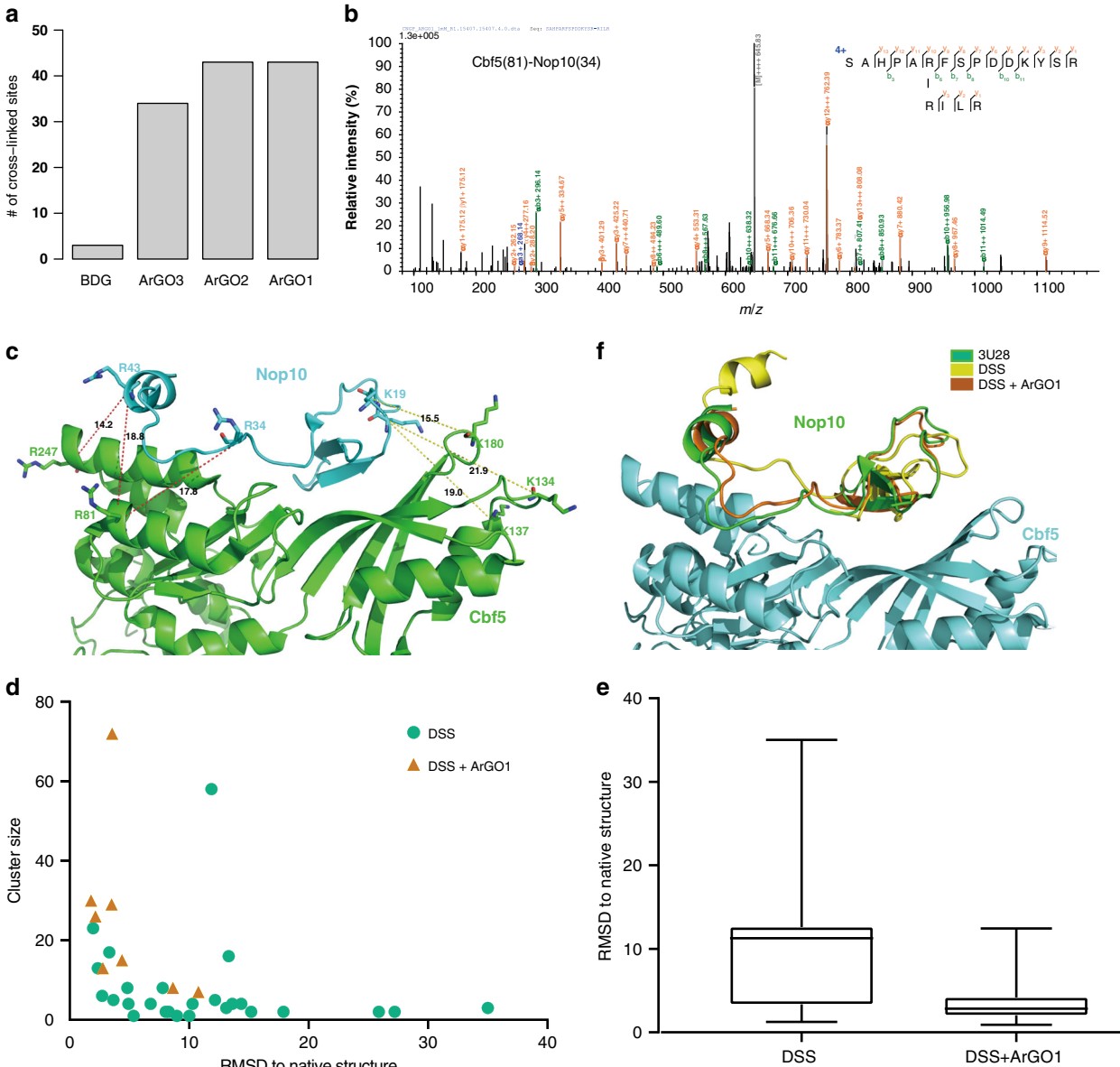

**Fig. 2** Cross-linking of the CNPG complex. **a** Number of cross-linked peptides identified from the reaction of ArGO1-3 or BDG with the CNGP complex. The CNGP complex was treated with 1 mM cross-linker at RT for 1 h. The identified cross-links using four cross-linkers are provided in the source data file. **b** Annotated MS/MS spectrum of an ArGO1-linked peptide pair from the CNGP complex. The precursor charge and *m/z* are shown in the spectrum. The highest peak (in bold) in the middle is the precursor. **c–f** Rosetta docking of Nop10 to Cbf5 with DSS and ArGO1 distance restraints. **c** Cross-links identified between Nop10 and Cbf5 are mapped to the yeast CNG sub-complex structure (PDB code: 3U28). Between Nop10 and Cbf5, three K–K cross-links were identified with DSS (dotted yellow lines) and three R–R cross-links were identified with ArGO1 (dotted red lines). **d** Summary of Rosetta docking results with DSS or DSS+ ArGO1 distance restraints. Each dot represents a cluster of conformations thus obtained, with the cluster size (number of poses in a cluster) and ligand−RMSD (the distance between a representative pose of a cluster and the native structure) shown on the y- and x-axis, respectively. **e** Box plot showing the distribution of the RMSD values of top 200 conformers obtained from Rosetta docking. The median, the interquartile range, the minimum, and maximum values are indicated by line, box, and whiskers (n = 200). The RMSD values of top 200 conformers obtained with Rosetta 3.10 using distance restraints of DSS cross-links alone or those of DSS+ ArGO1 are provided in the source data file. **f** Representative structure of the largest conformational cluster obtained with DSS or DSS+ ArGO1 cross-links, superimposed with the native structure

median, and highest number of KArGO cross-linked site pairs for any of the model proteins are 51, 81, and 102, respectively (Fig. 3d). These numbers are comparable to those obtained with DSS and BS[30]. The maximum $C_\alpha$–$C_\alpha$ distance of KArGO cross-links is calculated to be 32.2 Å, and 85.5% (324 out of 379) of the identified cross-linked K–R pairs fall within this limit according to the crystal structure of the model proteins (Fig. 3e). Notably, most of the KArGO cross-links are concentrated in the $C_\alpha$–$C_\alpha$ distance range of 8–28 Å (Fig. 3e).

**KArGO in structural analysis of protein complexes.** Once again, we used the CNGP complex to test the utility of KArGO in PPI analysis and modelling. KArGO produced 156 cross-linked K–R pairs, of which 119 are intra-protein cross-links and 37 are inter-proteins ones (Fig. 4a, Supplementary Table 2). An annotated MS/MS spectrum is shown in Fig. 4b. Compared to the ArGO and DSS results (Fig. 2c), KArGO cross-links yielded the most comprehensive surface coverage, as can be seen at the interface between Nop10 and Cbf5, and between Cbf5 and Gar1. Taking

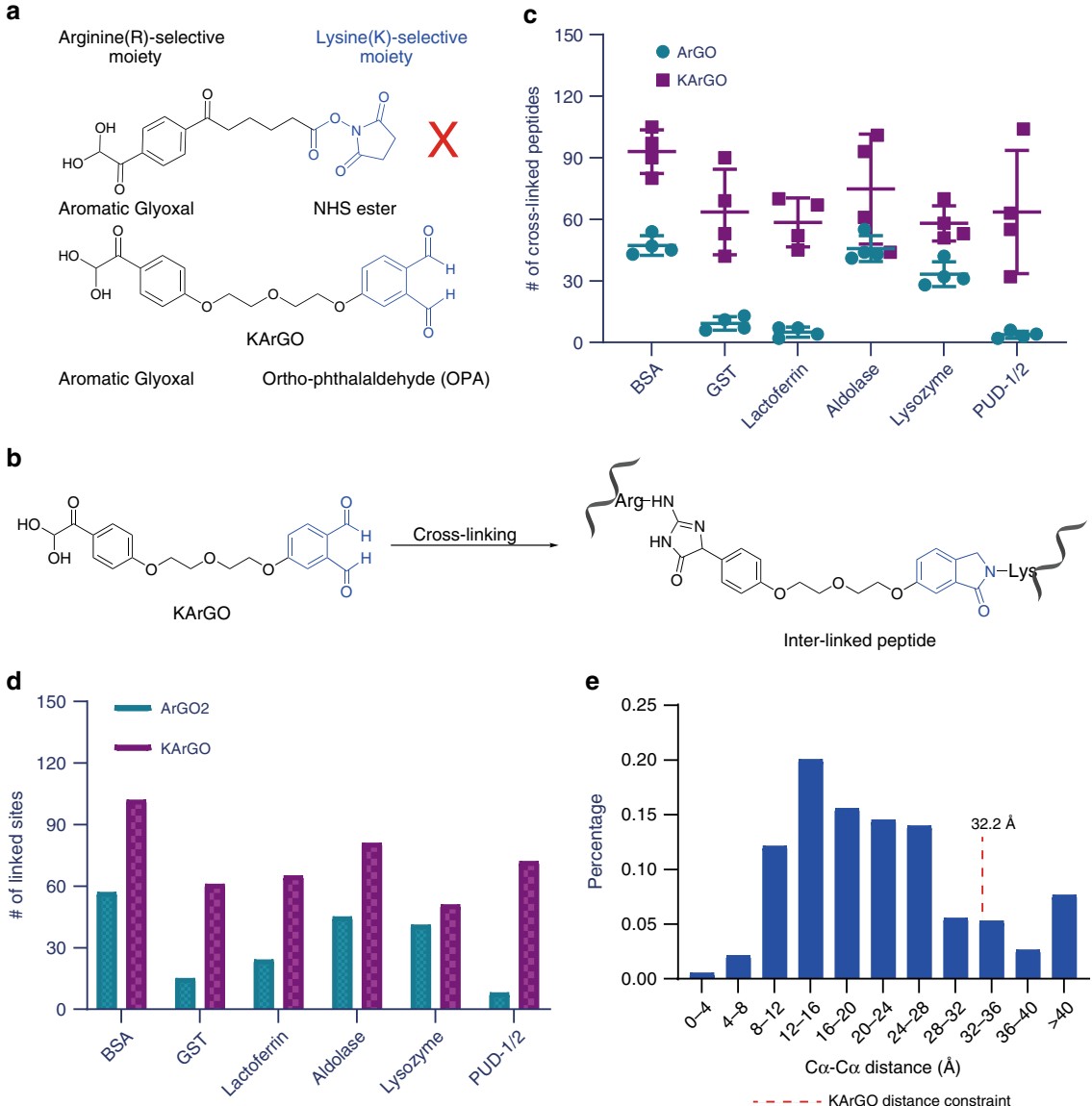

**Fig. 3** Development of KArGO, a hetero-bifunctional reagent for arginine-lysine cross-linking. **a** Structure of KArGO and an initially synthesised hetero-bifunctional cross-linker possessing ArGO and NHS functional groups. **b** Major cross-linked peptide structure identified from protein cross-linking with KArGO. The reaction of OPA with lysine residues forms a single phthalimidine product. **c** Comparison of the numbers of identified peptide pairs cross-linked by ArGO (blue dots) or by KArGO (purple squares) on model proteins. The four conditions of the ArGO experiment: 0.5 or 1.0 mM of either ArGO1 or ArGO2. The four conditions of the KArGO experiment: 0.1 or 0.2 mM of KArGO cross-link followed by trypsin or trypsin/Asp-N digestion. The derived mean and standard deviation (s.d.) values are shown in the figure. The number of ArGO- or KArGO-linked peptide pairs obtained from each of the four conditions can be found in the source data file. **d** The data in **c** are merged and reorganized by cross-linked R–R or K–R pairs. All cross-linked sites of ArGO2 and KArGO are provided in the source data file. **e** Histogram depicting the distribution of Cα–Cα distances of KArGO-linked residues, validated by comparison to the crystal structures of the model proteins. The structural compatibility rate, the proportion of cross-linked K–R pairs within the maximum distance restraint imposed by KArGO (32.2 Å), is 85.5%. See source data file for the calculated distance of each K–R pair

Nop10-Cbf5 as an example, five KArGO-linked K–R pairs are found along the length of the interface when mapped to the crystal structure of CNG: Nop10(K40)-Cbf5(R81) locks down the C-terminal region of Nop10, Nop10(R34)-Cbf5(K97) captures the unstructured central region of Nop10, and the remaining three cross-links pin down the N-terminal region of Nop10 relative to Cbf5 (Fig. 4c). In comparison, three K–K cross-links are found exclusively in the N-terminal region of Nop10, while three R–R cross-links are positioned at the C-terminal and central regions of Nop10 (Fig. 4c). Using the K–R distance restraints from KArGO cross-links in Rosetta docking, we quickly found that the largest conformational cluster had a predicted RMSD

value of less than 3 Å away from the crystal structure, which means that it is essentially not different from the native structure (Fig. 4d). In contrast, the largest conformational cluster obtained with the K–K (DSS) or R–R (ArGO) distance restraints had a predicted RMSD value greater than 10 Å. The structural model obtained by Rosetta docking using KArGO cross-links alone is as good as that obtained using DSS plus ArGO cross-links (Fig. 2d–f).

Additionally, we analysed the N-terminal domains (NTD) of alpha-kinase 1 (ALPK1) either alone or in complex with the ALPK1 kinase domain (KD)[60]. NTD and KD were expressed as separate recombinant proteins. ALPK1 is a 138.8 kDa protein

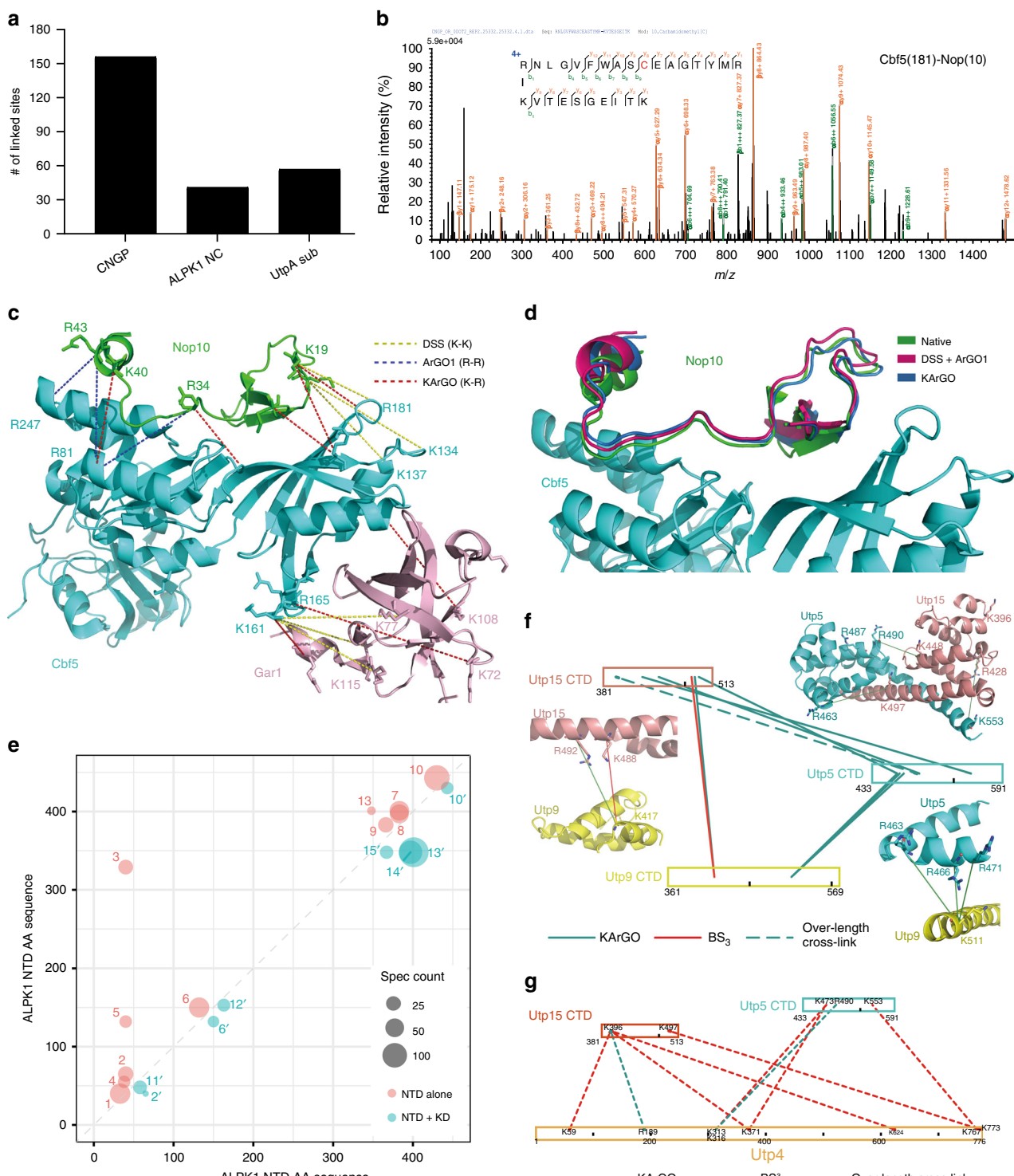

**Fig. 4** KArGO cross-linking of three protein complexes. **a** Number of KArGO-linked peptide pairs identified from each complex. **b** MS/MS spectrum of a pair of peptides cross-linked by KArGO from the CNGP complex. **c** Cross-links used for Rosetta docking of Nop10 to Cbf5, subunits of the CNGP complex. **d** Representative structure of Nop10 from the largest conformational cluster obtained after global docking with KArGO cross-links alone or with DSS plus ArGO1 cross-links, and superimposed on the native structure of the Nop10 subunit. **e** Amino-acid contact map depicting the K–R cross-links identified in ALPK1 NTD in the presence (blue) or absence (pink) of the ALPK1 kinase domain. Each circle represents a pair of KArGO-linked residues, marked with a number n or n′, which corresponds to the n[th] cross-link in Supplementary Table 3. The prime symbol denotes a cross-link originating from NTD + KD. The radius of each circle correlates with the number of MS2 spectra identified for this cross-link (see Supplementary Table 3). **f** Concerning Utp15, Utp9, and Utp5, three subunits of a UtpA sub-complex, KArGO cross-links (cyan) provided more information about the interface between subunits than BS[3] (red). The cross-links are mapped onto the cryo-EM structure[61] and also shown in a xiNET[67] connectivity map. A single over-length cross-link (dashed line) is also shown in the connectivity map. **g** A connectivity map showing that all of the high-confidence BS[3] (red) and KArGO (cyan) cross-links identified between Utp4 and Utp15 or Utp5 are over-length cross-links (dashed lines) when mapped onto the cryo-EM structure[61]

kinase harbouring a conserved alpha-kinase domain in the C-terminal region. A crystal structure of the NTD is available (PDB ID: 5Z2C)[60]. KArGO-based CXMS analysis identified 41 intra-domain and 3 inter-domain cross-links (Fig. 4a), of which 15 cross-links within NTD showed large change in spectral counts upon binding to KD (Fig. 4e, Supplementary Fig. 13, Supplementary Table 3). In particular, cross-links involving R38 or R40 (numbered 1-5 in Fig. 4e) diminished in the presence of KD. The spectral counts of R38 or K383 mono-links also decreased in the presence of KD, from 5 to 0 (R38) or from 1084 to 167 (K383). R40 mono-links were not detected under either condition. The above result suggests that binding of KD possibly buries R38/R40 and K383 of NTD.

Lastly, we tested KArGO with a UtpA sub-complex. UtpA consists of seven proteins (Utp4, 5, 8, 9, 10, 15, and 17) with a molecular weight of 648 kDa. Required for ribosome biogenesis, UtpA initiates pre-ribosome assembly by binding to nascent pre-rRNA and recruiting other factors that are necessary to process the pre-ribosomal particles[61]. Connectivity between the subunits of UtpA has been analysed by K–K cross-linking[61]. In the present work, we performed CXMS analyses of a UtpA sub-complex containing subunits Utp4, 5, 8, 9, and 15, and identified a total of 22 and 14 high-confidence (best $E$-value < 0.001, ≥6 spectra) inter-protein cross-links with BS[3] and KArGO, respectively (Fig. 4a and Supplementary Table 4). Mapping these inter-protein cross-links onto the cryo-EM structure of UtpA[61] we find that the K–R cross-links greatly complemented the K–K cross-links and provided rich information about the interface between subunits. For example, as shown in Fig. 4f, a single K–K cross-link, Utp15(K488)-Utp9(K417), was corroborated by a K–R cross-link Utp15(R492)-Utp9(K417). More importantly, seven other K–R cross-links depicted the interface of Utp15 and Utp5, and that of Utp5 and Utp9, which were not captured by BS[3] cross-linking. Seven out of the eight inter-protein K–R cross-links are consistent with the cryo-EM model of UtpA[61].

For protein–protein interactions involving Upt4, the KArGO cross-links complemented the BS[3] cross-links in a different way (Fig. 4g). Extensive K–K cross-links suggest that Utp4 is close to both Utp15 and Utp5, but they are all over-length cross-links when the distance was measured using the cryo-EM model. Two independently identified over-length K–R cross-links between Utp4 and Utp15 or Utp5 lend support to the K–K cross-links. These data strongly suggest that in the UtpA sub-complex analysed in this study, the Utp4 subunit takes a position that is closer to Utp15 and Utp5 than it does in the UtpA complex that had been analysed by cryo-EM.

## Discussion

In summary, we have developed a series of homo-bifunctional aromatic glyoxals (ArGOs) that cross-link proteins selectively at arginine residues; and a hetero-bifunctional cross-linker (KArGO) that targets both lysine and arginine. The structure of KArGO contains one aromatic glyoxal and one *ortho*-phthalaldehyde functional group, which is well known in amino-acid derivatization but previously unused in CXMS, for rapid and traceless conjugation of lysine residues. KArGO improves upon the ArGO cross-linkers, by enabling a shorter reaction time and lower reagent concentrations; and by offering greater protein surface coverage relative to homo-bifunctional cross-linkers. Furthermore, the performance of KArGO can exceed that of established lysine–lysine cross-linkers, such as DSS or BS[3] in some cases. For example, KArGO cross-links were identified across the entire length of the Nop10-Cbf5 interface of the yeast H/ACA complex, whereas DSS cross-links were concentrated in regions of higher lysine density. This resulted in convergence of

Rosetta models of the interface to within 3 Å of the native structure using KArGO restraints alone. Previously, our lab has demonstrated that the use of multiple cross-linkers such as K–K and K–C cross-linkers improves the depth of structural information obtained[30]. This is demonstrated again in this study: inter-subunit cross-links obtained with KArGO and BS[3] provide reinforcing evidence to locate the binding surface between subunits (UtpA) or domains (ALPK1), whereas cross-links unique to each offer structural information for distinct regions.

In addition to the K–C, R–R, and K–R cross-linkers, carboxylate–carboxylate (D/E–D/E, mainly) and zero-length amine-carboxylate (K–D/E, mainly) cross-linking reagents such as PDH[28], diazoker[29], EDC/NHS[27], and DMTMM[28] are also able to complement NHS ester-based K–K cross-linkers including BS[3] and DSS, which offer the most robust and reliable performance and thus are the staple in CXMS. Given unlimited amounts of samples and reagents, it would be desirable to use as many cross-linkers as possible to maximize structural coverage. However, in situations where one can only afford to try one or two non-NHS ester cross-linkers, considering the following four factors will be helpful.

First, the presence, distribution, and modification state (e.g. oxidation, acetylation, methylation, if known) of C, D, E, K, and R residues in the proteins of interest. This is usually an important issue for K–C cross-linking.

Second, pH and thermal stability of proteins, and potential conformational changes that might be induced when protein are shifted to a different pH or temperature, or by the addition of high concentrations of chemicals. This is relevant for EDC/NHS or DMTMM mediated zero-length K–D/E cross-linking and for DMTMM/PDH mediated D/E–D/E cross-linking. EDC prefers a rather acidic pH of 6.0[27], significantly away from the common pH range of 7.0–8.0 in CXMS practice. Cross-linking reactions involving DMTMM, including PDH and other dihydrazide reactions in which DMTMM pre-activates carboxylic acids, are conducted at 37 °C. If the reaction temperature is lowered to 25 °C, as in BS[3] or DSS cross-linking, the amounts of cross-linking products decrease significantly[28,29,62]. Conveniently, the ArGO and KArGO cross-linking conditions are nearly the same as that for BS[3] and DSS (pH 7–8, 25 °C), only with a supplement of 50 mM borate. Therefore, pH- or temperature-sensitive conformational changes are probably negligible between proteins samples treated with K–K, K–R, or R–R cross-linkers.

Third, structural compatibility of obtained cross-links with the crystal structures. Typically, more than 70% of K–K cross-links obtained with BS[3] and DSS are compatible with protein crystal structures[30], which is markedly higher than the compatibility rates of D/E–D/E (50–64%)[29] or zero-length K–D/E cross-links (33–41%)[28,30] (This is particularly concerning for zero-length cross-links because of their low structural compatibility rate, possibly having to do with their cross-linking pH or temperature. It remains unclear what distance restraints should be used for them in protein–protein docking or protein modeling, and why. In contrast, the R–R and K–R cross-links obtained with ArGO or KArGO have compatibility rates exceeding 80% (Supplementary Fig. 8).

Fourth, robustness of cross-linking chemistry, i.e. the number of cross-links that can be expected for an average protein. In this regard, zero-length K–D/E cross-linking is excellent, second only to K–K cross-linking by NHS esters. KArGO-mediated K–R cross-linking is reasonably robust, certainly better than R–R cross-linking, and possibly better than D/E–D/E cross-linking, which in one report generated only one cross-link for three model proteins each[28].

All considered, we think that the K–R cross-linking reagent KArGO is a highly competitive choice to complement NHS ester cross-linkers.

## Methods

**Reagents and protein solutions**. Tris(2-carboxyethyl) phosphine (TCEP) and 2-Iodoacetamide (IAA), were purchased from Pierce biotechnology (Thermo Scientific). HEPES, DMSO, NaCl, KCl, MgCl$_2$, Urea, CaCl$_2$, Methylamine, and Tris were purchased from Sigma. Boric acid was purchased from AMRESC. Acetonitrile (ACN), Formic acid (FA), Acetone, and NH$_4$HCO$_3$ were purchased from J. T. Baker. Mass-spectrometry-grade Trypsin and Asp-N were purchased from Promega.

**The synthesis of ArGOs and KArGO**. The synthetic procedures and NMR spectra of ArGO1-3, ArGO Analogues, BDG, and KArGO are provided in Supplementary Notes 5-14, Supplementary Figs. 15–102.

**Protein solutions**. Lyophilised proteins BSA, lysozyme, and lactoferrin were purchased from Sigma–Aldrich and dissolved in 20 mM HEPES, 150 mM NaCl at pH 7.4. An ammonium sulphate solution of aldolase was purchased from Sigma and dissolved in 20 mM HEPES, 150 mM NaCl at the required pH, and ammonium sulphate was removed using an Amicon filter.

**The purification of GST**. GST was purified from *E. coli* BL21 (DE3) by glutathione sepharose affinity chromatography and dissolved in 20 mM HEPES, 150 mM NaCl, pH 7.5 at 3 mg/ml.

**The purification of PUD-1/2 complex**. PUD-1 and PUD-2 were co-expressed at 16 °C for 16 h in the E coli BL21(DE3) strain (Novagen) and copurified as a dimer. The harvested cells were resuspended in buffer P500 (500 mM NaCl, 50 mM phosphate, pH 7.6,) and lysed using a high pressure cell disruptor (JNBIO) followed by sonication. The cell lysates were clarified by centrifugation and passage through a 0.45-μm filter and loaded onto a 5-ml HisTrap column (GE healthcare). After washed by 50 mM imidazole P500 buffer, the protein was eluted with 500 mM imidazole in P500 buffer. The PUD-1/2 complex was further purified using a Superdex 200 column (GE healthcare) equilibrated in buffer consisting 250 mM KCl, 5 mM HEPES-K, pH 7.6[63].

**The purification of ALPK1 NTD and KD complex**. To obtain well-behaved apo-ALPK1-(N + K) complexes, pGEX-6p-2 vector containing human ALPK1 (1–473) and pACSUMO-ALPK1 (959–1244) were transformed into *E. coli* strain BL21 (DE3) ΔhldE. When OD600 of cultured cells reached 0.8, 0.5 mM IPTG was added to induce protein expression at 20 °C for 16 h. The harvested cells resuspended in buffer containing 20 mM Tris-HCl (pH 8.0) and 500 mM NaCl, and then lysed with an ultrasonic cell disruptor. GST–ALPK1-(N + K) complexes were purified by glutathione sepharose affinity chromatography. GST was removed by overnight digestion with the homemade HRV 3 C protease at 4 °C. The supernatant containing the released apo-ALPK1-(N + K) was passed through fresh glutathione sepharose beads, further purified by gel filtration chromatography, and concentrated to 1 mg/ml[60].

**The purification of CNGP complex**. Cbf5, Nop10, and Gar1 were co-expressed in *E. coli* strain BL21(DE3) and mixed with separately expressed Nhp2 for complex formation[50]. The CNGP complex was copurified through HisTrap, heparin and gel filtration chromatography and dissolved in 20 mM HEPES-K (pH 8.0) and 500 mM NaCl.

**The purification of UtpA complex**. His$_6$-tagged Utp4 was expressed in *E. coli* BL21(DE3) strain and purified via HisTrap and heparin chromatography. The His$_6$-Smt3-tagged C-terminal domain (CTD) of Utp5 (residue 433-591) and the His$_6$-Smt3-tagged CTD of Utp15 (residue 380–513) were co-expressed and purified with a HisTrap column. After the His$_6$–Smt3 tags were cleaved with Ulp1, the complex was bound to a heparin column, eluted and passed through a HisTrap column to remove the cleaved His$_6$–Smt3 tag. The His$_6$–Smt3-tagged CTD of Utp8 CTD (residue 518–713) and the His$_6$–Smt3-tagged CTD of Utp9 (residue 361–569) were co-expressed and purified as above. To form the UtpA sub-complex, the purified Utp5–Utp15 CTD complex, Utp8–Utp9 CTD complex and His$_6$-tagged Utp4 were mixed and further purified via HisTrap and gel filtration chromatography. The UtpA sub-complex was finally dissolved in 10 mM HEPES-K (pH 8.0) and 200 mM NaCl.

**ArGO/KArGO stock solutions**. Twenty millimolar stock solutions of ArGO and KArGO were prepared in DMSO, and stored at −20 °C in a desiccant.

**Optimized ArGO protein cross-linking reaction conditions**. Twelve micrograms of protein (0.6 μg/μl) was cross-linked with 1 mM ArGO in a buffer of 50 mM borate, 50 mM HEPES (pH 7.5) at RT for 1 h. The reaction was quenched using 5× volumes of acetone for at least 30 min at −20 °C to precipitate the protein.

**Trypsin digestion and MS sample preparation**. The precipitated protein pellets were air dried and resuspended in 8 M urea, 100 mM Tris, pH 8.5. After reduction (5 mM TCEP, RT, 20 min) and alkylation (10 mM iodoacetamide, RT, 20 min), the samples were diluted 4-fold to 2 M urea using 100 mM Tris, pH 8.5. Denatured proteins were digested with trypsin (1:50 enzyme: substrate) for 2–4 h at 37 °C.

**Optimized KArGO protein cross-linking reaction conditions**. Twelve micrograms of protein (0.6 μg/μl) was cross-linked by 0.1/0.2 mM KArGO in a buffer mixture of 50 mM borate, 50 mM HEPES (pH 7.5) at RT for 15 min. The reaction was quenched using 5× volumes of acetone for at least 30 min at −20 °C to precipitate the protein.

**Trypsin and Asp-N digestion and MS sample preparation**. The precipitated proteins were air dried and resuspended in urea buffer as described above, before they were digested with trypsin (1:50 enzyme: substrate) at 37 °C for 4 h. Then, half of the sample was transferred to a new tube, to which Asp-N (1:100 enzyme: substrate) was added. Both digestions went on for another 12 h at 37 °C before LC-MS/MS analysis.

**Cross-linking of multiple protein complexes with KArGO**. CNGP complex or UtpA sub-complex, 10 μg (0.5 μg/μl) was cross-linked with 0.1, 0.2, and 0.4 mM KArGO in a buffer mixture of 50 mM borate, 50 mM HEPES (pH 7.5) at RT for 15 min. 12 μg (0.6 μg/μl). ALPK1 NTD or NTD-KD complex was cross-linked with 0.1, 0.2 mM KArGO at RT for 15 min. Then, 5× volume acetone was added to quench the reaction. Each sample was digested with trypsin alone and with trypsin plus Asp-N (see above) before LC-MS/MS analysis.

**Cross-linking of UTPA sub-complex with BS$^3$**. Ten micrograms of UtpA sub-complex (0.5 μg/μL) was cross-linked with 0.5 mM BS$^3$ at RT for 45 min, the reaction was quenched with 20 mM NH$_4$HCO$_3$.

**Mass spectrometry analysis**. The LC-MS/MS analysis was performed on an Easy-nLC 1000 II HPLC (Thermo Fisher Scientific) coupled to a Q-Exactive HF mass spectrometer (Thermo Fisher Scientific). Peptides were loaded on a pre-column (75 μm ID, 6 cm long, packed with ODS-AQ 120 Å–10 μm beads from YMC Co., Ltd.) and further separated on an analytical column (75 μm ID, 13 cm long, packed with Luna C18 1.9 μm 100 Å resin from Welch Materials) with a linear reverse-phase gradient from 100% buffer A (0.1% formic acid in H$_2$O) to 28% buffer B (0.1% formic acid in acetonitrile) in 56 min at a flow rate of 200 nL/min. The top 15 most intense precursor ions from each full scan (resolution 60,000) were isolated for HCD MS$^2$ (resolution 15,000; normalized collision energy 27%) with a dynamic exclusion time of 30 s. Precursors with unassigned charge states or charge states of 1+, 2+, >6+, were excluded.

**Identification of cross-links using pLink**. Both pLink1[8] and pLink2[46] software were used for cross-link identification. The pLink1 software was used to identify cross-linked peptides with precursor mass accuracy at 20 ppm, fragment ion mass accuracy at 20 ppm, and the results were filtered by applying a 5% FDR cutoff at the spectral level and then an E-value cutoff at 0.001[8].

The pLink2[46] software was used to identify cross-linked peptides with precursor mass accuracy at ±10 ppm, fragment ion mass accuracy at 20 ppm, and the results were filtered by applying a 5% FDR cutoff at the spectral level and then an SVM-score cutoff at 0.82. For the cross-linking condition optimization, we had a stricter filter condition—each cross-linked pair required two spectra. For the application of KArGO/ArGO to protein complexes, at least three spectra and best E-value < 0.001 were required. The search parameters used for pLink: instrument, HCD; precursor mass tolerance, 20 ppm; fragment mass tolerance, 20 ppm, the peptide length was set to 4–60, Carbamidomethyl[C] and Oxidation[M] as variable modification. The ΔM values for each ArGO/KArGO cross-linker are listed in Supplementary Table 1.

The identification results of ArGO and KArGO cross-links are provided in full in supplementary data 1 and supplementary data 2, respectively.

**Cα–Cα distance measurement**. The Cα–Cα Euclidean distances (ED) were measured using PyMOL in a PDB file, and the Solvent Accessible Surface Distance (SASD) was calculated using Xwalk. For BSA, GST, and Aldolase, the status of a cross-link (either intra- or inter-molecular) could not be determined based on the sequences of the cross-linked peptides. We thus calculated all the possible combinations and picked the ones with the shortest Cα–Cα distance. When calculating structure compatibility, the distance cutoffs are: 29.4 Å for ArGO1, 33.7 Å for ArGO2, 37.7 Å for ArGO3, and 32.2 Å for KArGO. The pdb files we use are as follows: BSA (3V03), GST (1Y6E), aldolase (3B8D), lysozyme (1LSY), lactoferrin (1FCK), PUD-1/2 (4JDE), CNG complex (3U28), ALPK-1 N-terminal domain with ADP-heptose (5Z2C), and UtpA sub-complex (5WLC).

**Rosetta docking**. Rosetta Version 3.10 and 3.5 were used to perform protein–protein docking simulations in Fig. 2 and Fig. 4, respectively. The ROSETTA flags are available in Supplementary Notes 2, 3, 4. The atomic structures of the CNG complex from PDB (code: 3u28), and the smaller partner Nop10 was treated as ligand. The pdb structure was prepacked and relaxed using the prepack and relax protocol, respectively. To save CPU time, only Cα was considered for ligand RMSD (L-RMSD) calculations in this study. In low resolution docking, 100,000 (for Nop10 to Cbf5-Gar1) models were generated. The 200 models with the lowest energy were clustered using an R script, as described previously[52]. The best model of each cluster was refined with the Rosetta local refinement protocol, and the pose with lowest energy was chosen as the representative model. Rosetta filtering was turned off to save CPU time, but additional filtering was performed optionally with an in-house Perl script, which required that the models satisfy all constraints given. Structures were illustrated using PyMOL.

The CXMS constraints were given in the following format: AtomPair {atom_name1} {residue_number1Chain_ID1} {atom_name2} {residue_number2Chain_ID2} BOUNDED {lb} {ub} {sd} {rswitch} {tag}. The parameters were lb = 0, ub = 24 (DSS), 29.4 (ArGO1), or 32.2 (KArGO), sd = 1 and rswitch = 0.5. The BOUNDED function is shown below[64,65]. The BOUNDED function is shown below[64,65].

$$f(x) = \begin{cases} 0, lb \leq x < ub \\ \left(\frac{x-ub}{sd}\right)^2, ub < x \leq ub + \text{rswitch} \cdot sd \\ \frac{1}{sd}\left(x - (ub + \text{rswitch} \cdot sd)\right) + \left(\frac{\text{rswitch} \cdot sd}{sd}\right)^2, x > ub + \text{rswitch} \cdot sd \end{cases} \quad (1)$$

**Reporting summary**. Further information on research design is available in the Nature Research Reporting Summary linked to this article.

## Data availability

The mass spectrometry raw data of ArGO and KArGO CXMS analysis have been deposited to the ProteomeXchange Consortium via the iProX partner repository with the dataset identifier PXD012341. The source data underlying Figs. 1c, d, 2a, e, 3c–e, and Supplementary Figs. 6, 7, and 12d are provided as a source data file. All other data are available from the corresponding authors on reasonable request.

## Code availability

All the software tools used in this study including pLink2 (version 2.3.2)[46], Xwalk (version 0.6)[66], and Rosetta (version 3.5 and 3.10)[52] are freely available, and the associated parameters or parameter files are provided in the Methods, Supplementary Methods, and Supplementary Notes 2-4.

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

## Acknowledgements

We thank Drs. Qiang Li and Dan Tan for helpful discussions, Maodong Li and Jincai Yang for calculating the length of the spacer arms of ArGO analogues, and Qiu-Yu Fu for advice in Rosetta Docking. We also thank Prof. Jiang Zhou for HRMS analysis. This work was supported by National Key Research and Development Program of China (2017YFA0505200), the Ministry of Science and Technology of China Projects 973 (2015CB856200 to X.L., 2014CB849800 to M.-Q.D., and 2013CB911203 to R.-X.S), the National Natural Science Foundation of China (21625201, 21661140001, 91853202, and 21521003 to X.L., and 21375010 to M.-Q.D.), the municipal government of Beijing, TIMBR, and Tsinghua University.

## Author contributions

X.L. and M.-Q.D. devised the project. A.X.J. designed and synthesised the compounds with assistance from Y.-L.T. and H.T. Y.C. performed the cross-linking experiments with assistance from Y.-H.D. and J.-H.W. Y.C., A.X.J., X.L., and M.-Q.D. analysed and interpreted the results of cross-linking experiments. Z.-L.C., R.-Q.F., J.Y. R.-X.S., and S.-M.H. provided assistance with the pLink software. R.-C.C., X.Z., and K.Y. provided the CNGP and UTPA protein complexes. N.H. aided analysis of Rosetta docking results. Y.S. and F.S. provided the ALPK1 protein. A.X.J., Y.C., X.L., and M.-Q.D. wrote the manuscript.

## Additional information

**Competing interests:** The authors declare no competing interests.

**Peer Review Information**: *Nature Communications* thanks Abdullah Kahraman, Michael Trnka and the other anonymous reviewer for their contribution to the peer review of this work. Peer reviewer reports are available.

