## [Peer Review File · Nature Communications]

Reviewers' comments:

Reviewer #1 (Remarks to the Author):

The authors have developed new homo- and heterobifunctional cross-linkers that are reactive towards lysine and arginine residues. The arginine-lysine cross-linker KArGo is particularly promising, as it provides complementary structural information to the conventional lys-lys cross-linker. The study is highly relevant for the structural proteomics community, however in its current state, many open questions and inaccurate descriptions make it difficult to judge the full impact of the work. These are my concerns in particular:

MINOR

- (p14, line 436): The sentence "In low resolution docking, 100,000 (for Nop10 to Cbf5-Gar1) models were generated. To save CPU time, only C α was considered protocol, and the smaller partner was treated as the ligand." is repeated twice in the same paragraph.

- Table S4b,c,d, lists information in the last column. What is this information. Please describe it in the table header.

- Please indicate the PDB ID in the header of Table S2-S4, which was used for the distance calculation.

Please also provide the PDB chain IDs of the proteins, which should help the reader to replicate your results with ease.

- You mention on page 5, line 160 that salt bridges of arginine might impair its reaction with ArGO. I can only agree with the authors. We saw a similar trend with DSS and the number of hydrogen bonds of cross-link resistant Lys residues (unpublished).

MAJOR

- Most cross-links that are mentioned in the paper text lack detailed information like in Table S2-4. Please provide similar tables for ALL cross-links in the paper, i.e. the 119 intra-protein cross-links from the CNGP complex using KArGO or the 34 intra-molecular and 21 inter-molecular cross-links from CNGP complex using ArGO1 and ArGO2 etc.

- Rather than using ROSETTA prepacking, please use the ROSETTA relax protocol, which is more relevant for a real case scenario. Prepacking relaxes the protein side chains only while keeping the main chain fixed in the bound state. In contrast, the relax protocol relaxes the main chain as well,

thereby creating a structure that is closer to the real case scenario of having available only unbound structures for protein-protein docking.

- Please provide all ROSETTA flags for all applied ROSETTA protocols. The options provided in the paper are incomplete, e.g. no options are provided for local docking, the nstruct and cst_file options are missing, etc.

- Can you elaborate to the reader what the mathematical background is for choosing the constraint function written on page 14.

- Can you justify why you used ROSETTA 3.5 for the docking calculations. It was published in 2013 and since has seen many improvements in terms of scoring function and sampling. Using the newest version might further improve your modelling results.

- To get a more accurate estimate of cross-links satisfying your distance constraints, please consider the computation of solvent accessible surface distances with tools like Jwalk and Xwalk. These tools are sensitive to protein curvatures and help to identify false positive cross-link identifications.

Reviewer #2 (Remarks to the Author):

The Jones et al. manuscript entitled “Improving Mass Spectrometry Analysis of Protein Structures with 1 Arginine-Selective Chemical Cross-linkers” describes the synthesis of novel cross-linkers towards arginine and lysine side chains. The manuscript is well written and it contains a lot of information for cross-linking community. However, the novelty of the manuscript is questionable. The glyoxal and ortho-phthalaldehyde concepts were introduced more than decade ago. Moreover, the current expansion of photo-inducible cross-linkers overcomes the limited selectivity problem. Even the manuscript represents a solid and comprehensive research it is not suitable for publication in Nature Communication. Also, there are some parts they are not clear or some statement are not referenced properly.

1) Line 40 – The first cross-linking paper is missing (Young et al, PNAS 2000).

2) Line 57 – The first paper describing carboxy-carboxy cross-linking is missing (Novak et al. EJMS 2007). Moreover, there is not any published article suggesting this chemistry inconvenient. It is not fair to say it does not work without any proof.

3) Line 124 – The cross-linking community agreed on cross-links nomenclature proposed by Sinz et al. JMS 2003 and Schilling et al. JASMS 2003). It must be corrected.

4) Line 128 - The final product represents only 20% yield. What’s the rest? Is it adjacent diol? If so, were the diol products analyzed as well? Can the diol undergo additional reactions? Also, it is necessary to specify the excess of bi-functional probe for 1c and d.

- 5) Line 155 – It would be beneficial to plot the number of theoretical distance constraints for the selected probe vs. identified cross-links. It is even more convenient to use Xwalk algorithm for such visualization.
- 6) Line 171 – Figure S10 is useless since the pH range 7.5-8.0 is very broad. It should be defined. Moreover, most protein complexes are intracellular where pH is between 6.5-7.0. It raises a question if the proposed chemistry is really beneficial and not counter-productive. It is mandatory to investigate proteins and protein complexes at their native environment. pH 8 is not considered native.
- 7) Line 177 – It looks like the product is unstable. Could you comment on that?
- 8) Line 243 – Why the reaction time for Argo is one hour and only 10 min for Kargo? Could you explain it?
- 9) Line 249 – Is it very difficult to judge if the number of cross-links favors Argo or Kargo since no information of all theoretical restrains is provided.
- 10) Line 254 – Could you comment what's the minimal restrain as well? 8A is really not so good.
- 11) The complete information of the synthesis is missing. What's the stock solution? Is it sensitive to moisture?
- 12) Line 377 and 386. It would be better to define the cross-linker over the protein ratio. High concentration of the cross-linker may introduce artifacts (Rozbesky et al. Anal.Chem. 2018). The gels in S6 suggest the over modifications.

Reviewer #3 (Remarks to the Author):

This manuscript from the Dong lab and co-workers introduces two new reagents for chemical crosslinking analysis based on arginine reactive aromatic glyoxal derivatives. The main claims of this paper are fairly straight forward: 1) chemistries that are orthogonal to the widely used Lys reactive reagents are necessary to improve the accuracy and precision of CLMS based structural modeling, 2) Aromatic glyoxal derivative reaction at arginine residues is an efficient and useful way to accomplish this. An additional advance here is the description ortho-phthalaldehydes as useful lysine directed crosslinking reagents. The Dong lab is one of premier groups focusing on crosslinking MS method development and analysis of multiprotein complexes.

It is clear that using alternative crosslinking reagents, other than the ubiquitous DSS or BS3 that modify amine-amine linkages, with either differing spacer arms or orthogonal specificity, will

improve CLMS guided structural biology. The same authors have a very nice paper (ref 28) in which they analyze the distribution of Lys-pairs, as well as Lys-Cys, and Lys-Asp/Glu pairs in experimental structures of multiprotein complexes deposited in the PDB. I recommend that the current manuscript extend this analysis to Lys-Arg and Arg-Arg linkages. The utility of these crosslinks in modeling is demonstrated here for a few examples, but it would be highly beneficial for someone designing a crosslinking protocol to have a sense of how generally Arg-reactivity is going to help, versus one of the other chemistries available. My bias is that the distribution of Lys and Arg in protein complexes are going to be quite similar (they are both positively charged and well distributed on the surface of complexes), so that if you were going to choose two reagents for a study it might be more useful to go with something else. That said, protein complexes are highly individualistic and it will always be helpful to have more option for crosslinking specificity.

Otherwise, I have two major concerns with the aromatic glyoxal chemistry. First, the current manuscript does not do enough to demonstrate the specificity of the reaction at Arg residues or characterize the reaction products in model systems very carefully. Figure S4 shows the results of reaction between ArGO and two model peptides. There are several deficiencies with this analysis. The model peptides encompass only a few nucleophiles and the product distributions of the reaction are not shown. Panel B shows a product ion spectrum corresponding to a precursor mass of the model peptide + the ArGO adduct, and yet no mass shifted product ions series is found in the spectrum. This indicates that the peptide has reacted and undergoes a neutral loss of the adduct during collisional activation. Hence, the ArGO adduct seems to be both non-specifically reacting as well as not stable, both of which are poor features for a crosslinker. Furthermore, the spectrum has a number of unannotated ion signals, especially for a model peptide reaction, that could indicate either a mixture of mono-ArGO adducts, or poorly characterized fragmentation of the adduct. Figure S3 illustrates heterogeneous reaction products including addition of a second molecule of glyoxal and interconversion between glycol and ketone forms of the adduct. This complicates MS-analysis as these species have different masses. More model peptides should be included, the full distributions of their products should be characterized with respect to all of the reaction outcomes noted in Figure S3.

Secondly, the authors find that the crosslinking products between the glyoxal derivatives and arginine are not stable. A slight change in the digestion time from 4-hours to overnight, results in loss of half of the crosslinks. While, they note this fact and develop methods to minimize storage and handling at room temperature, many crosslinking-MS labs are receiving samples from external collaborators and shipping and processing time makes it impossible to analyze the crosslinked samples immediately. The authors should characterize the stability of the crosslinked products more carefully and look at the rate of reversion, as this is a major practical impediment to widespread adoption of this method.

Minor:

Figure S1 – Figure is cropped so that names of the lower row compounds are not visible.

Figure 2 – The results of Rosetta docking of Nop10 to Cbf5 shows that the best DSS-only cluster outperforms any of the DSS+ArGO clusters in terms of RMSD to crystal structure ($\sim 2\text{\AA}$). When performing integrative modeling of protein complexes, the best solutions are not chosen solely based on cluster occupancy, but by other factors such as modeling scores, satisfaction of crosslinked restraints, recapitulation of known structural features that were not included in the model, etc. Therefore, the metric used here, that the addition of ArGO restraints increased the population of lower-RMSD clusters is not necessarily that useful if the correct answer can be derived from the DSG crosslinking. Figure 2e-f and show the three largest clusters, but not necessarily the three best clusters from each docking run.

Figure S12 – The alpha-kinase domain crosslinking shown here is fine in terms of demonstrating that the KArGO crosslinking can form crosslinks, but it doesn't really demonstrate utility over Lys-Lys crosslinking. Furthermore, the semi-quantitative comparison showing reduction of NTD crosslinks on addition of the KD, is not very carefully done in terms of demonstrating that the samples were normalized correctly, or that the reduction in crosslinking wasn't just due to the instrument not picking these peptide ions during DDA acquisition.

KArGO crosslinking alongside Trypsin/Asp-N digestion seems rather promising, and I am a little surprised that they didn't show more results from the UtpA complex, which seems like the most interesting system they tried.

Reviewers' comments:

Reviewer #1 (Remarks to the Author):

The authors have developed new homo- and heterobifunctional cross-linkers that are reactive towards lysine and arginine residues. The arginine-lysine cross-linker KArGo is particularly promising, as it provides complementary structural information to the conventional lys-lys cross-linker. The study is highly relevant for the structural proteomics community, however in its current state, many open questions and inaccurate descriptions make it difficult to judge the full impact of the work. These are my concerns in particular:

MINOR

- (p14, line 436): The sentence “In low resolution docking, 100,000 (for Nop10 to Cbf5-Gar1) models were generated. To save CPU time, only $C\alpha$ was considered protocol, and the smaller partner was treated as the ligand.” is repeated twice in the same paragraph.

Reply 1: Thank you for pointing it out. It has been corrected.

- Table S4b, c, d, lists information in the last column. What is this information. Please describe it in the table header.

Reply 2: The last column indicates whether or not the calculated distance between two cross-linked residues exceeds the upper limit of the given cross-linker. A header has been added to explain it.

- Please indicate the PDB ID in the header of Table S2-S4, which was used for the distance calculation.

Please also provide the PDB chain IDs of the proteins, which should help the reader to replicate your results with ease.

Reply 3: The PDB IDs and the chain IDs have been added.

- You mention on page 5, line 160 that salt bridges of arginine might impair its reaction with ArGO. I can only agree with the authors. We saw a similar trend with DSS and the number of hydrogen bonds of cross-link resistant Lys residues (unpublished).

MAJOR

- Most cross-links that are mentioned in the paper text lack detailed information like in Table S2-4. Please provide similar tables for ALL cross-links in the paper, i.e. the 119 intra-protein cross-links from the CNGP complex using KArGO or the 34 intra-molecular and 21 inter-molecular cross-links from CNGP complex using ArGO1 and ArGO2 etc.

Reply 4: We have added two supplementary Excel files (Tables S5 and S6) to provide such information.

- Rather than using ROSETTA prepacking, please use the ROSETTA relax protocol, which is more relevant for a real case scenario. Prepacking relaxes the protein side chains only while keeping the main chain fixed in the bound state. In contrast, the relax protocol relaxes the main chain as well, thereby creating a structure that is closer to the real case scenario of having available only unbound structures for protein-protein docking.

Reply 5: We repeated the modeling experiment using the newest version of ROSETTA

(ROSETTA 3.10) using the ROSETTA relax protocol, and we saw very similar results. We made sure that the two binding partners were separated, with the smaller one Nop10 treated as the ligand, before the backbones and the side chains were relaxed using the ROSETTA relax protocol.

- Please provide all ROSETTA flags for all applied ROSETTA protocols. The options provided in the paper are incomplete, e.g. no options are provided for local docking, the nstruct and cst file options are missing, etc.

Reply 6: All of the ROSETTA flags have been added to supplementary information and copied below. These flag files were used for both Rosetta 3.10 and 3.5 unless indicated otherwise.

Relax flag file (Rosetta 3.10 only):

```
-nstruct 1
-in:file:s combine.pdb
-relax:constrain_relax_to_start_coords
-relax:ramp_constraints false
-ex1
-ex2
-flip_HNQ
-no_optH false
-overwrite
```

Global docking flag file:

```
-s ../3u28_relaxed.pdb # input pdb name
-partners AB_C # docking partner
-dock_pert 3 8
-low_res_protocol_only
-randomize1
-randomize2
-spin
-constraints:cst_file ../LowRes_cng_KArGO
-no_filters
-run:debug
-run:use_time_as_seed true
-run:seed_offset 1 #add to random seed
-out:nstruct 500
-out:overwrite
-out:prefix sd1193_
-out:file:scorefile sd1193_LowRes_CNGP.fasc
-out:path:pdb ./
-out:path:score ./
```

cts file for DSS:

```
AtomPair CA 137A CA 18C BOUNDED 0 24 1 0.5 CNGP
AtomPair CA 134A CA 18C BOUNDED 0 24 1 0.5 CNGP
AtomPair CA 180A CA 19C BOUNDED 0 24 1 0.5 CNGP
```

cts file for DSS + ArGO

```
AtomPair CA 81A CA 34C BOUNDED 0 29.4 1 0.5 CNGP
AtomPair CA 81A CA 43C BOUNDED 0 29.4 1 0.5 CNGP
```

AtomPair	CA	247A	CA	43C	BOUNDED	0	29.4	1	0.5	CNGP
AtomPair	CA	137A	CA	18C	BOUNDED	0	24	1	0.5	CNGP
AtomPair	CA	134A	CA	18C	BOUNDED	0	24	1	0.5	CNGP
AtomPair	CA	180A	CA	19C	BOUNDED	0	24	1	0.5	CNGP

cts file for KArGO

AtomPair	CA	81A	CA	40C	BOUNDED	0	32.2	1	0.5	CNGP
AtomPair	CA	97A	CA	34C	BOUNDED	0	32.2	1	0.5	CNGP
AtomPair	CA	128A	CA	18C	BOUNDED	0	32.2	1	0.5	CNGP
AtomPair	CA	128A	CA	19C	BOUNDED	0	32.2	1	0.5	CNGP
AtomPair	CA	181A	CA	19C	BOUNDED	0	32.2	1	0.5	CNGP

- Can you elaborate to the reader what the mathematical background is for choosing the constraint function written on page 14.

Reply 7: The bounded constraint function in ROSETTA was applied as in previous studies (Demers *et al.*, *Nature Communications*, 2014 & Stephanie J. Hirst *et al.*, *Journal of Structural Biology*, 2011), where distance $\in [lb, ub]$, indicating that the cross-linker perfectly fits the model, the constraint force is set to zero without applying any penalty in scoring function; while distance $\in [ub, ub + rswitch \times sd]$, a harmonic penalty is applied; and when distance $> ub + rswitch \times sd$, the harmonic constraint is switched to a linear form to tolerate the large penalty.

$$f(x) = \begin{cases} 0, & lb \leq x < ub \\ \left(\frac{x - ub}{sd}\right)^2, & ub < x \leq ub + rswitch \cdot sd \\ \frac{1}{sd}(x - (ub + rswitch \cdot sd)) + \left(\frac{rswitch \cdot sd}{sd}\right)^2, & x > ub + rswitch \cdot sd \end{cases}$$

- Can you justify why you used ROSETTA 3.5 for the docking calculations. It was published in 2013 and since has seen many improvements in terms of scoring function and sampling. Using the newest version might further improve your modelling results.

Reply 8: We have repeated the docking experiment using ROSETTA 3.10 and arrived at the same conclusion. As can be seen in Fig. 2d (copied below), clustering of the top 200 models with the lowest energy (the best modeling score) clearly showed that, with additional help from the three ArGO1 cross-links, the Rosetta docking results converged much better. The top 200 models obtained with DSS+ArGO1 cross-links have in general much lower RMSD values than those obtained with DSS cross-links (Fig. 2d-e and below). The results demonstrate that the addition of ArGO constraint can greatly improve docking accuracy.

- To get a more accurate estimate of cross-links satisfying your distance constraints, please consider the computation of solvent accessible surface distances with tools like Jwalk and Xwalk. These tools are sensitive to protein curvatures and help to identify false positive cross-link identifications.

Reply 9: The newly added supplementary excel file contains the SASD information.

We thank the reviewer for careful reading of this manuscript and for the constructive comments and suggestions. We have learned new things in addressing the issues raised by the reviewer. Thanks very much for helping us improve this study!

Reviewer #2 (Remarks to the Author):

The Jones et al. manuscript entitled “Improving Mass Spectrometry Analysis of Protein Structures with 1 Arginine-Selective Chemical Cross-linkers” describes the synthesis of novel cross-linkers towards arginine and lysine side chains. The manuscript is well written and it contains a lot of information for cross-linking community. However, the novelty of the manuscript is questionable. The glyoxal and ortho-phthalaldehyde concepts were introduced more than decade ago. Moreover, the current expansion of photo-inducible cross-linkers overcomes the limited selectivity problem. Even the manuscript represents a solid and comprehensive research it is not suitable for publication in Nature Communication. Also, there are some parts they are not clear or some statement are not referenced properly.

Reply 10: Thank you for your feedback. Till now, CXMS is mostly limited to lysine specific cross-linkers, but they retrieve only a small portion of the structural information. Although photo-inducible cross-linkers can in theory react with any amino acid residue and thus overcome this limitation, identification of the resulting cross-linked peptides is still a challenge due to an explosion of search space and a lack of proper evaluation of search results. Our arginine selective cross-linkers are a timely contribution to the CXMS technology.

1) Line 40 – The first cross-linking paper is missing (Young et al, PNAS 2000).

Reply 11: The reference has been added to the manuscript.

2) Line 57 – The first paper describing carboxy-carboxy cross-linking is missing (Novak et al. EJMS 2007). Moreover, there is not any published article suggesting this chemistry inconvenient. It is not fair to say it does not work without any proof.

Reply 12: We looked up the said paper (Petr Novaka, and Anastassios E. Giannakopulosb, EJMS, 13, 105-113, 2007) but didn't find the description of the chemistry of carboxy-carboxy cross-linking reaction.

3) Line 124 – The cross-linking community agreed on cross-links nomenclature proposed by Sinz et al. JMS 2003 and Schilling et al. JASMS 2003). It must be corrected.

Reply 13: In the literature, dead end cross-link and mono-link, intrapeptide cross-link and loop-link are used interchangeably. We have added the alternative names in the revised manuscript. The term “interpeptide cross-link” and inter-link are not necessarily interchangeable. For example, just last year, Klykov et al. refer inter-links as a cross-link between two peptides from different proteins (Nature Protocol, 2018). To avoid confusion, in this manuscript we use cross-linked peptide or cross-link, both of which are commonly used in the literature, to denote “interpeptide cross-link,” which can be either within or between

proteins.

- 4) Line 128 - The final product represents only 20% yield. What's the rest? Is it adjacent diol? If so, were the diol products analyzed as well? Can the diol undergo additional reactions? Also, it is necessary to specify the excess of bi-functional probe for 1c and d.

Reply 14: We observed 81% conversion of limiting p-OMe-PhGO in the reaction with 2 equiv. N-acetyl arginine methyl ester. The low isolated yield of 3 is due to reversion to starting materials under the dilute conditions during purification. Diol 2 can undergo oxidation, or further conjugation with another molecule of ArGO to form product 6. Peptide conjugation experiments show these side-products to account for <10% of the product mixture (Fig. S5).

- 5) Line 155 – It would be beneficial to plot the number of theoretical distance constraints for the selected probe vs. identified cross-links. It is even more convenient to use Xwalk algorithm for such visualization.

Reply 15: The distance distributions of theoretically cross-linkable K-R, R-R pairs and identified crosslinks are shown below. This information has been incorporated into Fig. S8.

- 6) Line 171 – Figure S10 is useless since the pH range 7.5-8.0 is very broad. It should be defined. Moreover, most protein complexes are intracellular where pH is between 6.5-7.0. It raises a question if the proposed chemistry is really beneficial and not counter-productive. It is mandatory to investigate proteins and protein complexes at their native environment. pH 8 is not considered native.

Reply 16: Usually, the physiological pH refers to the pH of human blood, which is around 7.4, but the range of physiological pH is quite wide, as shown in the table below, which is taken

from a textbook. ArGO works similarly well at pH 7.0-8.0 (Fig. S6d). We typically use pH 7.5 for ArGO and KArGO cross-linking.

Table 3.1: pH values of various human body fluids and tissues.

Blood	7.35-7.45	Cerebrospinal fluid (CSF)	7.35-7.45
Gastric juice	1.0-2.0	Saliva	6.7-7.4
Bile	7.4-8.5	Tears	7.4
Human milk	6.9-7.0	Lysosomes	4.0-5.0
Mitochondria	6.6	Osteoblasts	9.0-10.0
Kupffer cells	6.4-6.5	Prostate	3.0-5.0
Urine	5.0-7.5	Stool	7.0-7.5

Textbook of Biochemistry with Biomedical Significance by Prem Prakash Gupta, CBS Publishers New Delhi, second edition

- 7) Line 177 – It looks like the product is unstable. Could you comment on that?
 Reply 17: The product 3 can convert back to 2 slowly (see Fig. 1c and Fig. S4). Evaluation of the stability of KArGO cross-links showed that KArGO cross-links are stable for at least one week (Fig. S12e) if cross-linked proteins are precipitated with acetone and stored as dried pellets at -80 °C, -20 °C or RT. This has been added to the revised manuscript (Fig. S12e).
- 8) Line 243 – Why the reaction time for Argo is one hour and only 10 min for Kargo? Could you explain it?
 Reply 18: The reaction between OPA and the amine group is very fast (in seconds), but the reaction between ArGO and arginine is slow, and the first step is reversible (Fig. 1a). ArGO cross-linking involves two slow reactions, whereas KArGO cross-linking is greatly facilitated by the instant reaction of OPA and amine. With one end attached to protein, the other end of KArGO is more likely to quickly find a local guanidine group to react with.
- 9) Line 249 – Is it very difficult to judge if the number of cross-links favors Argo or Kargo since no information of all theoretical restrains is provided.
 Reply 19: The theoretical restrains are described in the text (line 147 and line 253 in the original manuscript) and in Fig. S8. The max C α -C α distance is 33.7 Å for ArGO2, and 32.2 Å for KArGO.
- 10) Line 254 – Could you comment what's the minimal restrain as well? 8 Å is really not so good.
 Reply 20: The observed shortest C α -C α distance of KArGO's cross-links is 3.8 Å. For the majority of KArGO cross-links, the C α -C α distance falls within 8-28 Å.
- 11) The complete information of the synthesis is missing. What's the stock solution? Is it sensitive to moisture?
 Reply 21: The synthesis and spectroscopic data of ArGO/KArGO compounds are described in the SI p.26-29 (ArGO1-3), p. 36-46 (ArGO analogues) and p. 30-32 (KArGO). Typically, a 20 mM KArGO or ArGO stock solution is made in DMSO. KArGO and ArGO are both non-hydrolysable. Nonetheless, we store them at -20 °C in a desiccator.
- 12) Line 377 and 386. It would be better to define the cross-linker over the protein ratio. High concentration of the cross-linker may introduce artifacts (Rozbesky et al. Anal.Chem. 2018). The gels in S6 suggest the over modifications.
 Reply 22: The molar ratio of cross-linker over protein has a rather limited use. For example, a 1:50 (protein: cross-linker) ratio is not the same for a 10 kD vs a 100 kD protein. Usually the larger the protein, the more lysine and arginine residues it contains. For this reason, we think

that the mass ratio of cross-linker over protein is more informative—what works for a small protein is likely to work for a larger protein. The ArGO concentrations used in Fig. S6 ranged from 0.25 mM to 1.0 mM. For comparison, DSS or BS³ is typically used at 1.0 mM or above. The reviewer did not specify what in Fig. S6 suggests over-modifications. To our eyes, there was no sign of excessive cross-linking—there existed plenty of monomeric proteins after cross-linking and the dimeric or tetrameric bands all looked normal.

Reviewer #3 (Remarks to the Author):

This manuscript from the Dong lab and co-workers introduces two new reagents for chemical crosslinking analysis based on arginine reactive aromatic glyoxal derivatives. The main claims of this paper are fairly straight forward: 1) chemistries that are orthogonal to the widely used Lys reactive reagents are necessary to improve the accuracy and precision of CLMS based structural modeling, 2) Aromatic glyoxal derivative reaction at arginine residues is an efficient and useful way to accomplish this. An additional advance here is the description ortho-phthalaldehydes as useful lysine directed crosslinking reagents. The Dong lab is one of premier groups focusing on crosslinking MS method development and analysis of multiprotein complexes.

Thank you for the kind words.

It is clear that using alternative crosslinking reagents, other than the ubiquitous DSS or BS3 that modify amine-amine linkages, with either differing spacer arms or orthogonal specificity, will improve CLMS guided structural biology. The same authors have a very nice paper (ref 28) in which they analyze the distribution of Lys-pairs, as well as Lys-Cys, and Lys-Asp/Glu pairs in experimental structures of multiprotein complexes deposited in the PDB. I recommend that the current manuscript extend this analysis to Lys-Arg and Arg-Arg linkages. The utility of these crosslinks in modeling is demonstrated here for a few examples, but it would be highly beneficial for someone designing a crosslinking protocol to have a sense of how generally Arg-reactivity is going to help, versus one of the other chemistries available. My bias is that the distribution of Lys and Arg in protein complexes are going to be quite similar (they are both positively charged and well distributed on the surface of complexes), so that if you were going to choose two reagents for a study it might be more useful to go with something else. That said, protein complexes are highly individualistic and it will always be helpful to have more option for crosslinking specificity.

Reply 23: We analyzed the cross-linkable R-R, K-R pairs in 1808 complexes using Xwalk as described before (ref 29). By adding Arg-Arg or Lys-Arg cross-links, or both, the percentage of protein complexes with 5 or more virtual intermolecular cross-links increased from 67% to 82%, 86%, or 88%, respectively (distance ≤ 24 Å for all). In the category of protein complexes with more than 5 virtual intermolecular cross-links, Lys-Arg cross-linking is theoretically more prolific than Lys-Lys cross-linking. This has been added to the revised manuscript (Fig. S1). From this theoretical estimation and earlier experimental evidence, it is clear that arginine selective cross-linkers are a useful addition to CXMS.

Fig.

Fig. S1b

Otherwise, I have two major concerns with the aromatic glyoxal chemistry. First, the current manuscript does not do enough to demonstrate the specificity of the reaction at Arg residues or characterize the reaction products in model systems very carefully. Figure S4 shows the results of reaction between ArGO and two model peptides. There are several deficiencies with this analysis. The model peptides encompass only a few nucleophiles and the product distributions of the reaction are not shown. Panel B shows a product ion spectrum corresponding to a precursor mass of the model peptide + the ArGO adduct, and yet no mass shifted product ions series is found in the spectrum. This indicates that the peptide has reacted and undergoes a neutral loss of the adduct during collisional activation. Hence, the ArGO adduct seems to be both non-specifically reacting as well as not stable, both of which are poor features for a crosslinker. Furthermore, the spectrum has a number of unannotated ion signals, especially for a model peptide reaction, that could indicate either a mixture of mono-ArGO adducts, or poorly characterized fragmentation of the adduct. Figure S3 illustrates heterogeneous reaction products including addition of a second molecule of glyoxal and interconversion between glycol and ketone forms of the adduct. This complicates MS-analysis as these species have different masses. More model peptides should be included, the full distributions of their products should be characterized with respect to all of the reaction outcomes noted in Figure S3.

Reply 24: This is great advice. To address fully the specificity issue, we tested *p*-OMe phenyl glyoxal (OMe-PhGO) on seven synthetic peptides, covering all 20 amino acids (see Fig. S5 and below). The reaction products were analyzed by LC-MS/MS. The chromatographic peaks and the MS2 spectra of intact or modified peptides were analyzed manually, assisted by several software tools for spectrum labeling, de novo sequencing, expected or unexpected PTM search. We tried

our best to characterize and explain each product seen. We have verified that product 3 in Fig. S3 is the major product. We did find side products previously unknown to us (thanks to the reviewer again!), and two of them can be greatly reduced after TCEP treatment at 56 °C for 10 min, which is routinely done during sample preparation to reduce protein disulfide bonds. One of the side products, with a mass addition of 164.05 Da, seems to be a non-covalent conjugate between peptide and OMe-PhGO. The latter of which is readily lost as a neutral upon MS2, this side product becomes negligible after TCEP treatment. As to the concern that the ArGO2 modified LK-7 has a number of unannotated peaks in MS2. We found the major ones were internal ions containing an OMe-PhGO modified arginine. For example, m/z 665.305 is y6b5+, m/z 518.236 is y6b4+. The equivalent internal ions with unmodified arginine are also present in the MS2 spectrum of unmodified LK-7, with a mass reduction of 146.04 Da (below).

In summary, this set of experiment results have demonstrated that OMe-PhGO has good selectivity toward to arginine, and that the side products although exist in several forms, are not a big concern.

Table 1. Peptides tested

Peptide	Sequence	[M+H] ⁺	Containing R
HK-7	HPVCAYK	817.403	No
DK-10	DGMIKLWDLK	1218.655	No
VR-6	VKTELK	745.457	Yes
LK-7	LSQRFPK	875.510	Yes
FR-9	FVKQQWNLK	1218.674	Yes
SR-14	SDFKFSNLLGTVYK	1646.854	Yes
Ac-IR-7	Ac-IEAEKGR	844.452	Yes

Table 2. Distribution of peptide products after *p*-OMe-PhGO treatment.

(* incubated with 5 mM TCEP at 56 °C for 10 min after *p*-OMe-PhGO treatment)

Mass shift from intact peptide	Arginine-free peptide		Arginine-containing peptide					Nature of modified species	Mod. site as localized by MS2	Note
	HK-7	DK-10	VR-6 /VR-6 *	LK-7 /LK-7 *	FR-9	SR-14	Ac-IR-7			
+146.04 Da	-	-	1 /1	1 /0.97	1	1	1	Product 3 in Fig. S4	Arginine	
+162.03 Da	-	-	0.08 /0.12	0.08 /0.06	0.08	0.04	0.11	product of 2 in Fig. S4	Arginine	

+310.0 9 Da	-	-	0.08 /0.02	0.18 /0.06	-	-	-	Product 6 in Fig. S4	Arginine	
+164.0 5 Da	0.05	-	0.26 /0.02	0.37 /0.01	0.47	0.12	0.74	Non-covalent conjugate; p -OMe-PhG O readily dissociates as a neutral.	not on any particular residue; MS2 the same as that of the intact peptide	Greatly reduced after TCEP treatment
+134.0 4 Da	-	-	1.38 /0.74	0.37 /0.27	-	0.01	-	Not determined; a strong 1+ peak of 135.04 m/z in MS2	N-terminus	Elute after +134.04 on lysine; reduced after TCEP treatment
+134.0 4 Da	1	-	-	0.04 /0.03	0.01	-	0.15	Not determined; a strong 1+ peak of 135.04 m/z in MS2	lysine	Elute before +134.04 on the N-terminus
+27.99 Da	-	-	0.45 /0.36	0.17 /0.20	0.18	-	-	formylation	N-terminus	

c. MS/MS spectrum of unmodified LK-7

d. MS/MS spectrum of LK-7 + 146.04 Da

R*-NH3⁺ is the 1⁺ deaminated imonium ion of arginine to which p-OMe-PhGO is attached covalently.

Secondly, the authors find that the crosslinking products between the glyoxal derivatives and arginine are not stable. A slight change in the digestion time from 4-hours to overnight, results in loss of half of the crosslinks. While, they note this fact and develop methods to minimize storage and handling at room temperature, many crosslinking-MS labs are receiving samples from external collaborators and shipping and processing time makes it impossible to analyze the crosslinked samples immediately. The authors should characterize the stability of the crosslinked products more carefully and look at the rate of reversion, as this is a major practical impediment to widespread adoption of this method.

Reply 25: We designed a new experiment to test product stability. A mixture of BSA and Aldolase, 7.5 ug each, was cross-linked with 0.1 mM or 0.2 mM KArGO. Cross-linked proteins were precipitated by six volumes of acetone, and the resulting pellets were left at RT, 4 °C, or -20 °C for 1-7 days before they were stored to -80 °C till all the samples were collected for analysis. After digestion with trypsin or trypsin plus Asp-N, the samples were analyzed by LC-MS/MS. The results show that KArGO cross-links are stable for at least one week at RT, which is enough time for shipping of cross-linked samples.

Minor:

Figure S1 – Figure is cropped so that names of the lower row compounds are not visible.

Reply 26: It is an unnecessary line. The figure is not cropped.

Figure 2 – The results of Rosetta docking of Nop10 to Cbf5 shows that the best DSS-only cluster outperforms any of the DSS+ArGO clusters in terms of RMSD to crystal structure ($\sim 2\text{\AA}$). When performing integrative modeling of protein complexes, the best solutions are not chosen solely based on cluster occupancy, but by other factors such as modeling scores, satisfaction of crosslinked restraints, recapitulation of known structural features that were not included in the model, etc. Therefore, the metric used here, that the addition of ArGO restraints increased the population of lower-RMSD clusters is not necessarily that useful if the correct answer can be derived from the DSG crosslinking. Figure 2e-f and show the three largest clusters, but not necessarily the three best clusters from each docking run.

Reply 27: See Reply 8 to reviewer 1. We agree that cluster occupancy and other factors such as modeling scores should all be taken into consideration. When the largest clusters and the best scoring clusters do not agree with each other, it can be difficult to judge which ones are better than the others. We repeated the docking experiment using ROSETTA 3.10, which is a more up-to-date version. The results are shown in Fig. 2d-f. With DSS+ArGO1 cross-links, the largest clusters also have the lowest RMSD values relative to the crystal structure. The winner is clearly DSS+ArGO1.

Figure S12 – The alpha-kinase domain crosslinking shown here is fine in terms of demonstrating that the KArGO crosslinking can form crosslinks, but it doesn't really demonstrate utility over Lys-Lys crosslinking. Furthermore, the semi-quantitative comparison showing reduction of NTD crosslinks on addition of the KD, is not very carefully done in terms of demonstrating that the samples were normalized correctly, or that the reduction in crosslinking wasn't just due to the instrument not picking these peptide ions during DDA acquisition.

Reply 28: The Lys-Lys cross-linking results had been published. Yes, the conformational change can be spotted from the Lys-Lys cross-links. The Lys-Arg cross-links are a confirmation of that. The spectra count of the linear peptides of ALPK1 NTD in the presence or absence of KD is 11005 or 10279, respectively. These two numbers are very close to each other (1.07: 1), suggesting that there was no difference in protein quantity in the two samples.

KArGO crosslinking alongside Trypsin/Asp-N digestion seems rather promising, and I am a little surprised that they didn't show more results from the UtpA complex, which seems like the most interesting system they tried.

REVIEWERS' COMMENTS:

Reviewer #1 (Remarks to the Author):

Thanks for addressing all my comments. Just a couple of remarks:

- The sentence "In low resolution docking, 100,000 (for Nop10 to Cbf5-Gar1) models were generated." is still repeated; now on page 15.
- Please mention in your main text that you also run the relax protocol, which gave similar results to the prepack protocol. You can use your rebuttal response as a template.
- Would be nice if you could do a similar plot like Figure S8 for SASD distances and comment on it in your main text.

Reviewer #2 (Remarks to the Author):

According to point-by-point response, the authors answered the majority of concerns raised by reviewers adequately. However, there are still few details that need to be addressed prior publishing. Even the use of photo-inducible cross-linker is not straightforward due to data mining it gives the highest number of distance restraints so far. Further, the alternative carboxy-carboxy or zero-length cross-linking strategy (incorrectly reference in the manuscript, it should be both 10.1255/ejms.963 and 10.1073/pnas.1320298111) generates additional constrains that can be used for Rosetta simulation as well. It is quite a misfortune, the manuscript reports the comparison of traditional Lys-Lys chemistry to newly develop one (Arg-Arg and Lys-Arg) only. It is correct the pH varies according the source of material. However, the majority of studied proteins are intracellular where the pH is slightly below 7. If presented technique works at pH 7.5 it would be better to be more accurate in the text instead using range 7-8. It might be possible to get slightly different outcome of the reaction if it is done at pH 7 or 8 due to the increase concentration of protons (pH 7) favoring the dehydration of vicinal diols. In order to improve the manuscript both issues should be discussed.

Reviewer #3 (Remarks to the Author):

The authors have responded rigorously to all of the reviewer comments, and I support publication of this paper with no further revisions.

REVIEWERS' COMMENTS:

Reviewer #1 (Remarks to the Author):

Thanks for addressing all my comments. Just a couple of remarks:

- The sentence "In low resolution docking, 100,000 (for Nop10 to Cbf5-Gar1) models were generated." is still repeated; now on page 15.

Reply 1: Sentence is no longer repeated (line 490).

- Please mention in your main text that you also run the relax protocol, which gave similar results to the prepack protocol. You can use your rebuttal response as a template.

Reply 2: We mention this in line 224.

- Would be nice if you could do a similar plot like Figure S8 for SASD distances and comment on it in your main text.

Reply 3: We have added plot of SASD distance of ArGO and KArGO to Supplementary Figure 7b. The structural compatibility rates dropped to around 58% by SASD for ArGO2 and KArGO.

Reviewer #2 (Remarks to the Author):

According to point-by-point response, the authors answered the majority of concerns raised by reviewers adequately. However, there are still few details that need to be addressed prior publishing. Even the use of photo-inducible cross-linker is not straightforward due to data mining it gives the highest number of distance restrains so far. Further, the alternative carboxy-carboxy or zero-length cross-linking strategy (incorrectly reference in the manuscript, it should be both 10.1255/ejms.963 and 10.1073/pnas.1320298111) generates additional constrains that can be used for Rosetta simulation as well. It is quite a misfortune, the manuscript reports the comparison of traditional Lys-Lys chemistry to newly develop one (Arg-Arg and Lys-Arg) only. It is correct the pH varies according the source of material. However, the majority of studied proteins are intracellular where the pH is slightly below 7. If presented technique works at pH 7.5 it would be better to be more accurate in the text instead using range 7-8. It might be possible to get slightly different outcome of the reaction if it is done at pH 7 or 8 due to the increase concentration of protons (pH 7) favoring the dehydration of vicinal diols. In order to improve the manuscript both issues should be discussed.

Reply 4: We have added a paragraph in the introduction which addresses the advantages and disadvantages of photo-inducible cross-linkers:

In theory, non-selective cross-linkers such as glutaraldehyde and photo-induced cross-linkers can retrieve more structural information than residue-selective cross-linkers, because they can react with most if not all amino acid residues. However, identification of the resulting cross-linked peptides is a huge challenge due to an explosion of the search space and a lack of proper evaluation of identification results.

We have also added the following text to the discussion, which discusses the merits of KArGO within the context of cross-linker selection:

In addition to the K-C, R-R, and K-R cross-linkers, carboxylate-carboxylate (D/E-D/E, mainly) and zero-length amine-carboxylate (K-D/E, mainly) cross-linking reagents such as PDH,²⁸ diazoker,²⁹ EDC/NHS,²⁷ and DMTMM²⁸ are also able to complement NHS ester-based K-K cross-linkers including BS³ and DSS, which offer the most robust and reliable performance and thus are the staple in CXMS. Given unlimited amounts of samples and reagents, it would be desirable to use as many cross-linkers as possible to maximize structural coverage. However, in situations where one can only afford to try one or two non-NHS ester cross-linkers, considering the following four factors will be helpful.

First, the presence, distribution, and modification state (e.g. oxidation, acetylation, methylation, if known) of C, D, E, K, and R residues in the proteins of interest. This is usually an important issue for K-C cross-linking.

Second, pH and thermal stability of proteins, and potential conformational changes that might be induced when protein are shifted to a different pH or temperature, or by the addition of high concentrations of chemicals. This is relevant for EDC/NHS or DMTMM mediated zero-length K-D/E cross-linking and for DMTMM/PDH mediated D/E-D/E cross-linking. EDC prefers a rather acidic pH of 6.0,²⁷ significantly away from the common pH range of 7.0-8.0 in CXMS practice. Cross-linking reactions involving DMTMM, including PDH and other dihydrazide reactions in which DMTMM pre-activates carboxylic acids, are conducted at 37 °C. If the reaction temperature is lowered to 25 °C, as in BS³ or DSS cross-linking, the amounts of cross-linking products decrease significantly.^{28, 29, 63} Conveniently, the ArGO and KArGO cross-linking conditions are nearly the same as that for BS³ and DSS (pH7-8, 25 °C), only with a supplement of 50 mM borate. Therefore, pH- or temperature-sensitive conformational changes are probably negligible between proteins samples treated with K-K, K-R, or R-R cross-linkers.

Third, structural compatibility of obtained cross-links with the crystal structures. Typically, more than 70% of K-K cross-links obtained with BS³ and DSS are compatible with protein crystal structures,³⁰ which is markedly higher than the compatibility rates of D/E-D/E (50-64%)²⁹ or zero-length K-D/E cross-links (33-41%)^{28, 30} (. This is particularly concerning for zero-length cross-links because of their low structural compatibility rate, possibly having to do with their cross-linking pH or temperature. It remains unclear what distance restraints should be used for them in protein-protein docking or protein modeling, and why. In contrast, the R-R and K-R cross-links obtained with ArGO or KArGO have compatibility rates exceeding 80% (Supplementary Figure 8).

Fourth, robustness of cross-linking chemistry, i.e. the number of cross-links that can be expected for an average protein. In this regard, zero-length K-D/E cross-linking is excellent, second only to K-K cross-linking by NHS esters. KArGO mediated K-R cross-linking is reasonably robust, certainly better than R-R cross-linking, and possibly better than D/E-D/E cross-linking, which in one report generated only one cross-link for three model proteins each.²⁸

All considered, we think that the K-R cross-linking reagent KArGO is a highly competitive choice to complement NHS ester cross-linkers.

Reviewer #3 (Remarks to the Author):

The authors have responded rigorously to all of the reviewer comments, and I support publication of this paper with no further revisions.

Reply5: Thank you very much for your positive comments.